# Mental navigation in the primate entorhinal cortex

Sujaya Neupane[1], Ila Fiete[1,2] & Mehrdad Jazayeri[1,2]✉

A cognitive map is a suitably structured representation that enables novel computations using previous experience; for example, planning a new route in a familiar space[1]. Work in mammals has found direct evidence for such representations in the presence of exogenous sensory inputs in both spatial[2,3] and non-spatial domains[4–10]. Here we tested a foundational postulate of the original cognitive map theory[1,11]: that cognitive maps support endogenous computations without external input. We recorded from the entorhinal cortex of monkeys in a mental navigation task that required the monkeys to use a joystick to produce one-dimensional vectors between pairs of visual landmarks without seeing the intermediate landmarks. The ability of the monkeys to perform the task and generalize to new pairs indicated that they relied on a structured representation of the landmarks. Task-modulated neurons exhibited periodicity and ramping that matched the temporal structure of the landmarks and showed signatures of continuous attractor networks[12,13]. A continuous attractor network model of path integration[14] augmented with a Hebbian-like learning mechanism provided an explanation of how the system could endogenously recall landmarks. The model also made an unexpected prediction that endogenous landmarks transiently slow path integration, reset the dynamics and thereby reduce variability. This prediction was borne out in a reanalysis of firing rate variability and behaviour. Our findings link the structured patterns of activity in the entorhinal cortex to the endogenous recruitment of a cognitive map during mental navigation.

A hallmark of cognition is the ability to organize experiences into knowledge that can be retrieved flexibly to perform novel mental computations. One way the mammalian brain solves this problem is by establishing cognitive maps that encode spatial, temporal and other abstract relationships in the environment[1,11,15]. The representational building blocks of cognitive maps have been extensively studied in spatial contexts. For example, sensory experiences during both physical and virtual navigation can drive spatially selective responses in the hippocampus and the entorhinal cortex (EC)[2,3,16–19]. Non-spatial variables such as temporal relations[9,20,21], value[4], social hierarchy[10], memory traces[22–24] and abstract stimuli[5–8] can also evoke neural responses that reflect the underlying relational structures.

A crucial prediction of the cognitive map hypothesis is that the brain can exploit the structure of the latent map in the absence of sensory inputs to perform purely mental computations[1,11,25,26]. To test this idea, we designed a mental navigation task (MNAV) for monkeys in which they used a joystick to move at a constant speed between designated start and target positions along a horizontal line punctuated by six equidistant landmarks, which we refer to as the landmark line (Fig. 1a). Two monkeys were at first trained to use the joystick to navigate between a subset of landmark pairs, with all landmarks visible. Subsequently, all landmarks were invisible during movement. As such, the monkeys had to compute and produce displacement vectors without sensory feedback.

We first familiarized the monkeys with the task, the landmark line and joystick use through a navigate-to-sample task (NTS) (Extended Data Fig. 1a). For each trial, after fixating a central spot, the monkeys were presented with a target landmark below the fixation point and the landmark line above the fixation point. After the fixation point changed colour ('Go' cue), monkeys could deflect the joystick to translate the entire landmark line horizontally in either direction. They received a reward for releasing the joystick when the landmark above the fixation point matched the target landmark below (Supplementary Video 1). The start and target landmarks were chosen randomly in every trial. While performing the NTS task, the monkeys gained experience with the relative positions of landmarks and the joystick's speed—the two key variables needed for solving the main MNAV task.

After performance in the NTS task reached a criterion (see Methods and Extended Data Fig. 1b), the monkeys were introduced to the MNAV task (Fig. 1a). MNAV is similar to NTS in that the monkeys use the joystick to move along the landmark line to arrive at the target. However, MNAV differs from NTS in that neither before nor after the onset of joystick deflection can the monkeys see the landmark line. Before joystick deflection, only the start landmark above the fixation point is visible. During joystick movement, all landmarks above the fixation point are invisible. These modifications force the monkeys to solve the MNAV task using their memory of relative landmark positions rather than direct visual input. After the joystick offset, the landmark

[1]McGovern Institute for Brain Research, Massachusetts Institute of Technology, Cambridge, MA, USA. [2]Department of Brain and Cognitive Sciences, Massachusetts Institute of Technology, Cambridge, MA, USA. ✉e-mail: mjaz@mit.edu

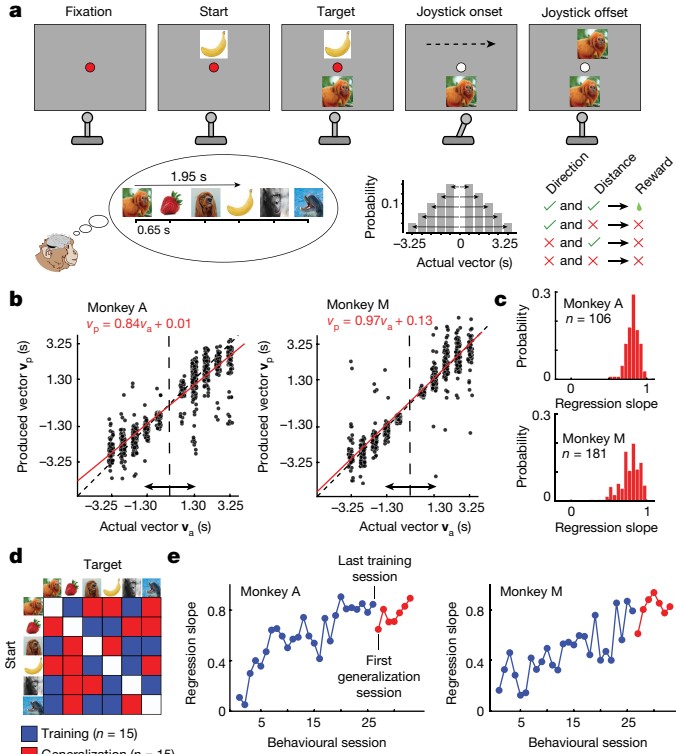

**Fig. 1 | Behavioural task, performance and generalization. a**, MNAV task. Top, an example trial. Left to right: The monkey fixates on a central fixation point. The start and target landmarks are presented sequentially above and below the fixation point, respectively. When the colour of the fixation point changes, the monkey must use the joystick to produce a vector from the start to the target landmark. As soon as the joystick is deflected (onset), the area above the fixation becomes blank and remains so until the joystick offset. During joystick deflection, landmarks move, invisible to the monkey, with a fixed speed (black dashed arrow). After the joystick offset, the landmark closest to the fixation point is revealed. Bottom left, six equidistant landmarks with a fixed ordinal position. Adjacent landmarks are 0.65 s apart. The monkey must learn the sequence and use memory to produce the correct vector by deflecting the joystick in the correct direction and for the appropriate duration of time. In the example, the target landmark (tamarin monkey) is three images to the right of the start landmark (banana). Therefore, the monkey must deflect the joystick rightward for 1.95 s ($3 \times 0.65$). Bottom middle, distribution of directions and distances for all start–target pairs ($6 \times 5$ conditions). Bottom right, reward contingencies (see Methods). **b**, The produced vector ($\mathbf{v}_p$) as a function of the actual vector ($\mathbf{v}_a$) for every trial (black circles) in two representative sessions. Performance was quantified by the regression line (red) relating $\mathbf{v}_p$ to $\mathbf{v}_a$. **c**, Distribution of regression slope across all behavioural sessions. **d**, Start and target landmark pairs used during training (blue) and generalization (red). **e**, Learning trajectory on the training pairs (blue) and on the held-out generalization pairs after the performance on the training pairs had stabilized (regression slope $\geq 0.8$, $P < .0001$ at 95% confidence interval; see Methods).

closest to the current centre position in the landmark line is revealed, and a reward is provided if it matches the target landmark. If not, the monkey is given a second and final chance to make a corrective movement and receive a smaller reward (in our analyses, we treated trials with a second attempt as error; see Methods). These modifications make MNAV a purely mental navigation task in which monkeys have to deflect the joystick in the correct direction and for the correct duration to travel between landmarks without any sensory feedback (Supplementary Video 2). The monkeys learned to perform MNAV (Fig. 1b,c). The produced vectors ($\mathbf{v}_p$) quantified in terms of temporal distance (magnitude) and direction (sign) closely matched the actual vectors separating the start and target landmarks ($\mathbf{v}_a$).

In principle, MNAV could be solved with two different strategies. One strategy is to treat the task as a stimulus–response memorization problem with a look-up table for each of the 30 (6 choose 2) possible pairs of landmarks and a desired vector. This 'model-free' strategy does not require the monkeys to learn the structured relationships between landmarks. An alternative 'model-based' strategy is to learn and rely on this structure to produce the vectors. The latter strategy involves more sophisticated learning but reduces memory load and offers flexibility when faced with new conditions (for example, a previously untraversed start–target pair). To evaluate the monkeys' strategy, from the outset, we divided the 30 pairs into 2 appropriately balanced (in terms of direction and distance) disjoint sets, 15 training conditions and 15 held-out conditions to assess their ability to generalize (Fig. 1d). We reasoned that if the monkeys use a model-free strategy, learning the training set would not confer the knowledge needed to solve the generalization set without additional experience. By contrast, having learned the structure should allow immediate generalization. Evaluating the monkeys' performance with this paradigm, we found that they were able to readily generalize with high performance from the very first session (Fig. 1e). This finding suggests that the monkeys solved the MNAV task using knowledge about the structure of the landmark line.

The EC encodes spatial displacements[3], responds to spatial and non-spatial variables[5,7,17], has been hypothesized to support vector-based path integration and navigation[14,27] and receives sensory input about spatial landmarks[28]. Accordingly, we hypothesized that the EC might have a central role in mental navigation. We recorded spiking activity in the EC during MNAV and focused our analysis on the vector production epoch during joystick movement (Fig. 2). In both monkeys, task-modulated neurons were concentrated in a small region in the posterior EC (Extended Data Fig. 1c–e and Supplementary Tables 1 and 2). Three prominent features were evident in the activity profile of task-modulated neurons. In some neurons, firing rates were punctuated by transient bumps (Fig. 2a,b). In others, firing rates ramped and reached different levels depending on distance (Fig. 2a). There were also neurons with a combination of ramping and transient bumps (Fig. 2a). Finally, in many neurons, there was a strong increase in activity before joystick offset.

Considering that landmarks were invisible during navigation, and that no other sensory feedback was provided, the presence of transient bumps in subsets of neurons is striking. We hypothesized that the bumps are endogenously generated activity modulations associated with the memorized relative position of landmarks. This hypothesis predicts that the time between consecutive bumps should be 0.65 s, the same as the temporal distance between landmarks. To test this prediction, we computed the autocorrelogram (ACG) of spiking activity for each trial, averaged ACGs across trials and estimated the time lag associated with the peak of the first side lobe. We considered a neuron to have periodic activity if the value of the ACG at the peak, denoted as the periodicity index (PI), was two standard deviations above the mean of the null distribution (see Methods and Fig. 2c).

Across sessions, the proportion of neurons with a significant PI ranged from 0% to 45% depending on the recording site in the posterior EC, with an overall average of 37% (231/614) and 36% (311/864) in monkeys A and M, respectively. These percentages are comparable to the percentage of distance-tuned cells in the medial EC (MEC) in mice running on a treadmill in the dark[29], and to the percentage of grid cells (GCs) in candidate regions of the rodent MEC[3,30], but higher than the percentage of GCs reported in primates[17] during a free-viewing paradigm. Of note, across neurons with significant PI, the peak lag was at or near 0.65 s, which matches the temporal distance between consecutive landmarks (Fig. 2d and Extended Data Fig. 2a). Complementary analyses revealed no such periodicity in the monkeys' eye and hand (joystick) movements (Extended Data Fig. 3), which suggests that neural signals were generated endogenously and were associated with the memory of the landmark line. This finding provides compelling evidence that the

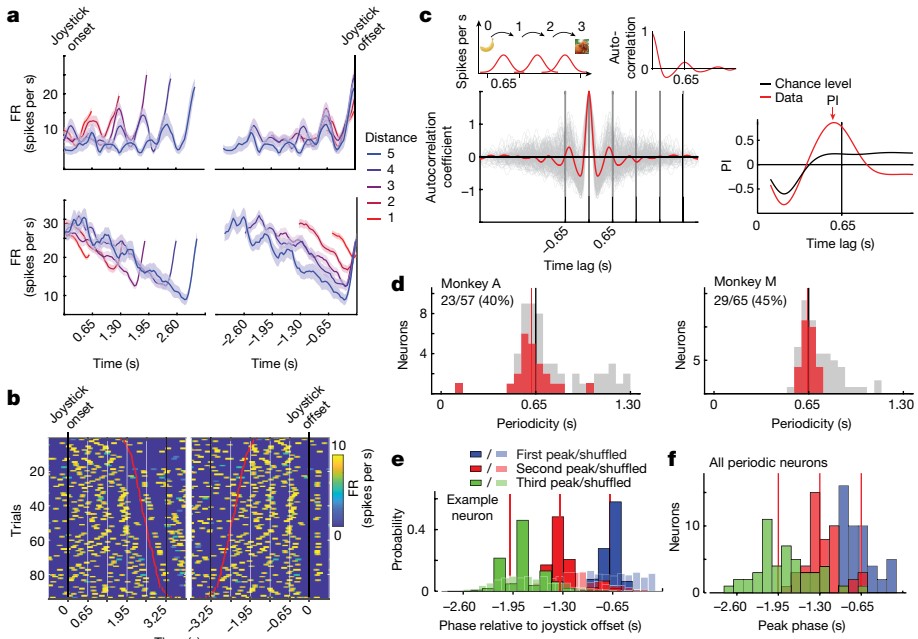

**Fig. 2 | Neural signatures of mental navigation in the EC. a**, Firing rate (FR) of two neurons aligned to joystick onset (left) and joystick offset (right), colour-coded by temporal distance. Lighter wide lines represent s.e.m. **b**, Raster plot of the firing rate of a neuron for trials with a temporal distance of higher than 3, aligned to joystick onset (left) and joystick offset (right). Trials are sorted in ascending order of the monkey's joystick offset times, denoted by red dots. **c**, Top, schematic showing how localized activity peaks at landmarks (left) would lead to an autocorrelation function with peaks at multiples of 0.65 s (right). Bottom left, single-trial (grey) and average (red) ACG of the neuron shown in **b**. Bottom right, PI at various time lags for the neuron in **b**. PI was considered significant when its value was above the chance level, defined as two standard deviations above the PI derived from surrogate Gaussian Process data (black, see Methods). **d**, Periodicity values of recorded neurons in one session (red, significant PI; grey, not significant). See Extended Data Fig. 3 for periodicity across all neurons. **e**, Phase distribution of the first (blue), second (red) and third (green) firing rate local maxima before joystick offset. The first peak was estimated within a window of 1,000 ms before joystick offset, the second peak within a window of 1,000 ms before the first peak and the third peak within a window of 1,000 ms before the second peak. Phase distributions were calculated on the basis of the mean firing rate bootstrapped over 100 subsamples of trials. The corresponding null distributions were obtained from shuffled spike trains (KS test, $P \ll 0.0001$). **f**, Distribution of the first (blue), second (red) and third (green) peak phase before joystick offset across neurons.

EC had a representation of the temporal structure needed for mental navigation between landmarks.

Next, we sought to test the hypothesis that the observed periodicity is linked to behaviour. One prediction of this hypothesis is that periodicity should be relatively weak or absent outside the mental navigation epoch. To test this prediction, we compared the PI at a 0.65-s period between the mental navigation epoch and the inter-trial interval (ITI). The PI was significantly stronger during the mental navigation epoch (Extended Data Fig. 2b), which suggests that the periodic activity was specific to mental navigation.

Another prediction of this hypothesis is that the trial-by-trial variability in the periodic activity should correspond to variability in behaviour. We tested this prediction in three ways. First, we compared the PI between correct (rewarded) and incorrect (unrewarded) trials. Neurons that were significantly periodic within a 100-ms window centred at 0.65 s had a lower PI during error trials (Extended Data Fig. 2c), suggesting that weaker periodicity contributed to committing larger errors. Second, we asked whether joystick offset time was correlated with phase lag of the periodic activity on a trial-by-trial basis. To address this question, we developed an analysis to quantify the phase associated with local activity peaks relative to the time of joystick offset (see Methods). The first peak preceding the joystick offset was centred close to −0.65 s and differed significantly from a null distribution generated by applying random phase shifts to the same spiking data (KS test, $P \ll 0.0001$) (Fig. 2e). We extended this analysis to earlier times in the navigation epoch and found that these were centred near multiples of 0.65 s (Fig. 2e,f and Extended Data Fig. 4a,b). This finding complements the autocorrelogram analysis and further validates the presence of a structured relationship between local peaks and the temporal structure of landmarks. Finally, we examined the relationship between behavioural variability and periodicity, asking whether over-shooting (or undershooting) the target was associated with increased (or decreased) periodicity in task-modulated neurons. To address this question, we pooled rewarded trials associated with the same desired vector displacement, sorted them on the basis of monkeys' produced vector magnitude and compared the average periodicity between the top and bottom tertiles. We found a small but significant increase in the periodicity (that is, lower frequency) for the top tertile compared to the bottom tertile, suggesting that fluctuations of periodicity might contribute to behavioural variability (Extended Data Fig. 4c,d). Together, these results establish a close link between periodicity in the EC and mental navigation, suggesting that the transient bumps in the firing of neurons in the EC helped the monkeys to track their position on the landmark line.

The other common feature of EC responses during MNAV was ramping activity. Ramping activity has long been associated with timing[31–34]. In general, three aspects of the ramp can encode a target time interval: the ramp's initial state, its slope and its end state. Accordingly, we characterized the coding properties of ramping activity across our EC population. We first focused our analysis of onset versus offset coding (Fig. 3a). Using linear regression, we quantified the relationship between temporal distance and firing rate estimated from spiking data in a 100-ms window before joystick onset and offset (Fig. 3b). Some neurons encoded distance at joystick onset, others at joystick offset, and a small number of neurons had a representation of distance at both onset and offset (Fig. 3b; see Methods). We also performed a

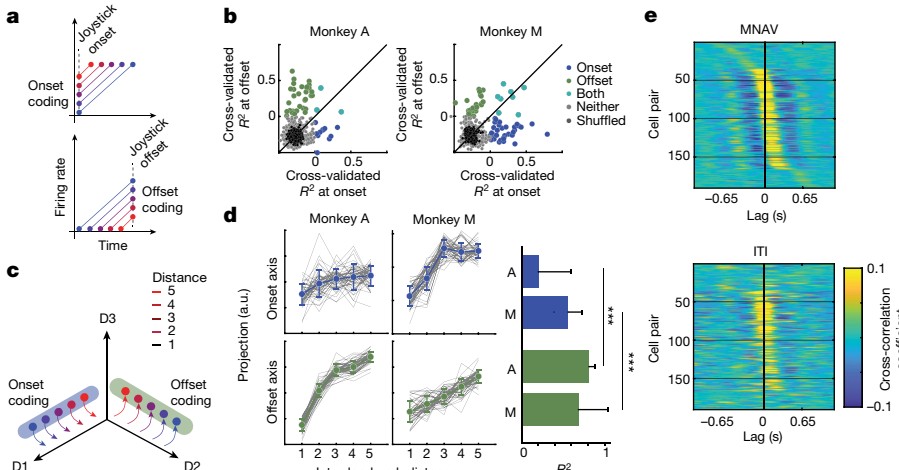

**Fig. 3 | Temporal distance coding and signatures of attractor dynamics in the EC. a**, Schematic depiction of how single neurons might encode temporal distance at the joystick onset time (top) or joystick offset time (bottom) and track elapsed time by monotonic change of firing rates. **b**, Scatter plot showing cross-validated explained variance ($R^2$) for a linear regression model at onset (abscissa) and offset (ordinate). Different colours represent $R^2$ for subsets of neurons with positive temporal distance coding at onset only (blue), at offset only (green), at both onset and offset (turquoise), with no positive coding (light grey) and for a null distribution generated from randomly shuffled regressors (black). **c**, Schematic depiction of how the neural states across the population of neurons might encode temporal distance at onset (blue axis) and/or offset (green axis). **d**, Left, projection of held-out neural states onto the axes associated with onset (top, blue) and offset (bottom, green) coding estimated using targeted dimensionality reduction. Right, goodness-of-fit ($R^2$) for onset and offset coding for the two monkeys (A and M) (two-sided Wilcoxon rank-sum test; $n = 50$ bootstraps, $Z = -7.81$, $P \ll 0.0001$ for monkey A and $n = 50$ bootstraps, $Z = -4.17$, $P \ll 0.0001$ for monkey M). a.u., arbitrary units. **e**, Cross-correlation structure of 20 simultaneously recorded periodic neurons with the highest periodicity metric (188 pairs) rank ordered on the basis of activity during the navigation epoch (top: MNAV) and plotted with the same order for the ITI (bottom) for monkey A (see Extended Data Fig. 7 for monkey M).

complementary targeted dimensionality reduction analysis to quantify onset and offset coding at the population level (Fig. 3c). Similar to the single-neuron analysis, we found cross-validated encoding axes for distance at both joystick onset and joystick offset (Fig. 3d). However, the encoding of distance at joystick onset was weak and/or present only for a subset of distances (Fig. 3d, blue). By contrast, population neural states in both monkeys had a robust linear relationship with distance before joystick offset (Fig. 3d, green). These results reveal two complementary representations of the target interval (or distance) in the EC: one that represents the desired distance before navigation has begun (across initial states) and another that tracks the distance travelled during navigation (across terminal states). Both signals might have a role in navigation. The former might initialize the dynamics at the start of navigation, and the latter track the current state during navigation. The presence of strong offset coding in the EC differs qualitatively from ramping activity in other areas of the brain that are associated with classical time interval production tasks. In timing tasks that do not involve mental navigation, the target interval is usually encoded by the initial state and the slope, and not by the end state when ramping activity reaches a common threshold[32,33]. Our finding suggests that neural computations that support mental navigation through time are distinct from those that support producing scalar time intervals with no intervening landmarks.

Next, we compared the encoding properties of the slope of ramping to those of the firing rates. Specifically, we quantified the degree to which the target interval is explained by the variance of firing rate or the slope of firing rate within a 300-ms window before the joystick offset (−550 ms to −250 ms relative to joystick offset). A direct comparison of the two indicated that the distance effect due to the firing rate is significantly larger than that due to the slope of the firing rate (Extended Data Fig. 4e). This result further highlights the distinct signatures of mental navigation through time, as compared wth classical timing tasks in which adjustments of slope have a central role[33]. Together, these results suggest that the main feature of ramping activity in the EC is an encoding of temporal distance near the joystick offset.

In our experiment, distances are not fully independent of landmark identities. For example, the end landmarks are the only ones that partake in the longest distances, and the intermediate ones never do so. As such, the presence of distance coding in the EC might be a by-product of neurons' selectivity to specific landmarks. To distinguish between distance coding and landmark selectivity, we performed a four-way analysis of variance (ANOVA) with distance, start landmark, target landmark and direction as factors. We found that significantly less variance of the neural firing rate near joystick offset was explained by either the start or the target landmark compared with distance (Extended Data Fig. 4f,g). This finding is consistent with the EC encoding distances invariant to individual landmarks—a more abstract and generalizable representation of the task structure.

The presence of endogenously generated periodic activity in task-modulated EC neurons is consistent with the behaviour of a continuous attractor network (CAN)[12–14]. Accordingly, we asked whether periodic EC neurons in our population exhibit features expected by attractor dynamics. A key feature of CAN dynamics is that the distribution of relative firing phases across cell pairs is conserved across conditions[12,13]. To test this prediction in our dataset, we measured pairwise correlations between simultaneously recorded EC neurons with the highest PI in two contexts—during mental navigation and during ITIs. We found that the distribution of relative phases between cell pairs was conserved across the MNAV and ITI conditions across 190 unique cell pairs (Fig. 3e and Extended Data Fig. 5a–d). Notably, the structure of correlations across the pairs was similar between the two periods (monkey A: $r(188) = 0.84$, $P \ll 0.0001$; monkey M: $r(292) = 0.86$, $P \ll 0.0001$). This finding was robust and generalized to all task epochs, including the inference epoch (during start and target landmark presentation but before mental navigation), error trials and for both directions of movement (Extended Data Fig. 5a–d).

These results raise two key questions. First, what circuit mechanisms enable the EC to learn and recall the temporal structure of external landmarks? Second, what are the implications of this recall for behaviour? To address the first question, we adapted a CAN model developed for

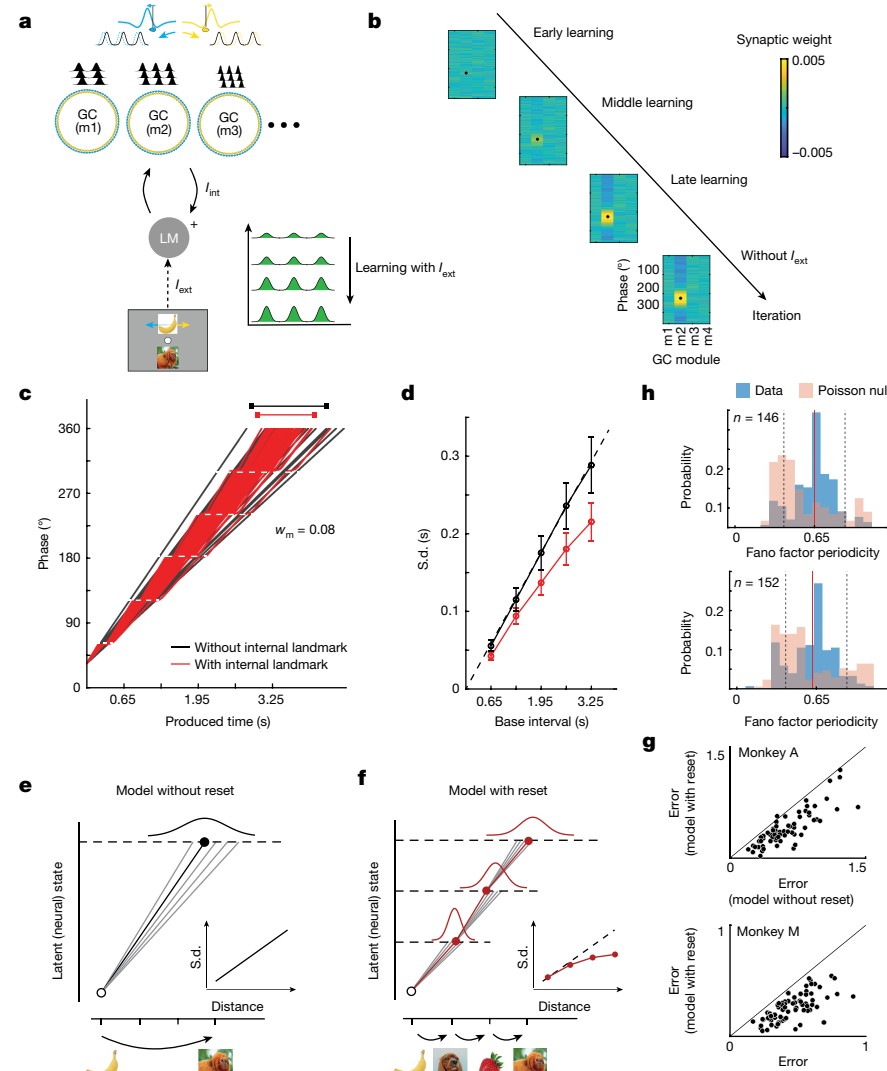

**Fig. 4 | The behaviour of the monkeys is consistent with the dynamics of a CAN model. a**, Model schematic. A LM driven by external landmarks ($I_{ext}$) interacts bidirectionally with grid-cell (GC) modules (m1, m2, ...). The GC to LM input ($I_{int}$) has plastic synapses ('+'). GC modules integrate motion through asymmetric centre-surround connectivity and velocity inputs (top). **b**, During visual navigation, GC–LM connections change. The synaptic drive from GC cells whose periodicity and phase match that of $I_{ext}$ is gradually strengthened (black dot: $I_{ext}$ periodicity and phase). The network maintains its selectivity after $I_{ext}$ is removed. **c**, Network state trajectory across 50 simulations under noisy velocity input, with (red) and without (black) landmark inputs (dotted white lines: reset events due to endogenous landmarks). **d**, Standard deviation (s.d.) grows linearly with temporal distance in the absence of landmark inputs

and sublinearly in their presence (model: s.d. = $a \times$ mean$^b$ + $c$; H0: $b$ = 1; H1: $b$ < 1; one-tailed $t$-test(999) = −11.29, $P \ll 0.0001$). **e,f**, Two models of behavioural variability. In the model without reset (**e**), the s.d. increases linearly (**e**, inset). In the model with reset (**f**), it grows sublinearly (**f**, inset). Consequently, the distribution of produced temporal distances is wider for the no-reset model (top black Gaussian) than the reset model (top red Gaussian). **g**, The model with reset (ordinate) provides a better fit to the monkey's behaviour compared with that without reset (abscissa) (paired $t$-test, monkey A: $t$(77) = 7.93, $P \ll 0.0001$, monkey M: $t$(101) = 16.56, $P \ll 0.0001$). **h**, Distribution of Fano factor periodicity of neurons with a significant Fano factor PI compared with their corresponding (Poisson) null data. Dotted lines denote the window in which the significance of the Fano PI was tested. See Extended Data Fig. 8 for details.

GCs in rodents during navigation[14] to model neural dynamics during MNAV. The CAN model consists of neurons with lateral interactions that lead to a pattern of repeating bumps. These bumps move in the presence of velocity inputs, thereby supporting path integration. Moreover, inputs associated with external landmarks can reduce errors accumulated during path integration[35–38].

To adapt the CAN model to MNAV, we made three assumptions. First, we assumed that either efference copy and/or reafference during joystick deflections provide the velocity input to update the CAN state, as has been suggested for path integration in abstract spaces[5]. Second, we assumed that model units interact with 'landmark' neurons (LMs) that relay information about external landmarks when they are present, which is consistent with how landmark information is thought to

influence EC activity[28,39]. Finally, we assumed that connections between CAN and LM units are plastic, which is a key component of learning cognitive maps[27,40]. We reasoned that, during NTS, this plasticity might strengthen connections between LM activity and CAN units with matching dynamics, thereby allowing CAN to reproduce LM activity during MNAV, when landmarks are invisible. We tested this idea using a simple model in which an LM unit receives inputs from external landmarks as well as from CAN units associated with different phases and periodicities (that is, analogous to GC modules in rodents) through plastic synapses (Fig. 4a). In the presence of external landmarks, synapses were modified such that CAN inputs mimicked the external input, and when the external input was extinguished, the CAN drive alone was able to emulate that external drive (Fig. 4b and Extended Data Fig. 6a,b). For the

specific parameters of the MNAV task (0.65 s between landmarks), the model was able to drive LM with the subset of CAN units whose phase and frequency matched the external temporal structure (Fig. 4b). This simple model provides a candidate circuit mechanism for the storage of external landmarks in the EC.

This model raises a question about the source of cell–cell correlation patterns in the data (Fig. 3e). In general, we suggest three possibilities. First, the periodic neurons could be homologous to landmark cells recorded in rodents during navigation. Second, our periodic task-modulated neurons might be homologous to GC cells in rodents. The presence of endogenous periodicity and the preserved cell–cell correlations are more consistent with the GC interpretation than with the former hypothesis. However, we found a significantly tighter clustering of the relative phase distribution of cell–cell cross-correlation in our data, which differs from what is predicted for GCs in rodents, making the GC interpretation inconclusive. Finally, the periodic neurons in our population could be neither landmark cells nor GCs, but, instead, part of an attractor network within the functional architecture of the EC that is capable of producing memory traces for the landmarks[22–24] (see Supplementary Information for a detailed discussion).

Next, we asked how the endogenous landmarks influence model behaviour during mental navigation. To address this question, we compared two models—one with the ability to learn and reconstruct landmarks endogenously and one without it. The model that did not learn landmarks performed path integration by integrating noisy velocity inputs (Fig. 4c, black) and generated temporal distances whose standard deviation grew linearly with the base interval (Fig. 4d, black), mimicking the well-known scalar property of interval timing[41,42]. By contrast, the dynamics of the model that learned to generate landmarks endogenously were punctuated with reset-like events coincident with the timing of landmarks (Fig. 4c, red). In other words, each endogenous landmark temporarily 'pinned' the active bumps and slowed down the movement of the network pattern, acting as a transient reset for the dynamics. This reset mechanism can be readily explained by how LM input interacts with CAN. The LM input provides local stability to CAN states, making it less responsive to velocity input.

In spatial navigation, external landmarks are thought to act as anchor points that help to reduce navigation errors[14,35,36,39]. An intriguing hypothesis is that the endogenous landmarks serve a similar error-correcting function. If so, it is expected that the variability would reduce periodically at the reset events during mental navigation. To test this possibility, we measured the Fano factor (that is, the variance divided by the mean) across model units as a function of time during the navigation epoch. As expected by the presence of reset events, the Fano factor of individual units fluctuated periodically with a periodicity of 0.65 s. Further model simulations indicated that the periodicity of the Fano factor closely followed the periodicity of unit activations across a range of speeds (Extended Data Fig. 6d).

The observation of a periodic Fano factor in the model motivated a similar analysis of neural activity in the EC. Notably, many EC neurons had a periodic Fano factor that was tightly clustered around 0.65 s (Extended Data Fig. 8a,d). This periodicity was not a by-product of the underlying periodicity of firing rates, and was also evident in neurons with non-periodic activity (Extended Data Fig. 8c,f). The proportion of neurons with a periodic Fano factor was 43% (378/864) in monkey M and 55% (341/614) in monkey A. These results highlight the possibility of an error-reducing process associated with the activation of the endogenous landmarks across the EC network.

Next, we sought to examine the effects of this error-reducing scheme on behavioural variability. In classical time interval production tasks, behaviour is characterized by scalar variability[41,42]—that is, variability whose standard deviation scales with the base interval. This was also true in the behaviour of the CAN model in the absence of landmarks (Fig. 4d, black). By contrast, the model with the endogenous landmark input exhibited subscalar variability—that is, variability whose standard deviation grew sublinearly with the base interval (Fig. 4d, red). We verified this sublinearity by nonlinear regression in which the standard deviation and mean had a power-law relationship with an exponent of less than unity (s.d. = $a \times$ mean$^b$ + $c$; H0: $b$ = 1; H1: $b$ < 1; one-tailed $t$-test(999) = −11.29, $P$ << 0.0001). This relationship is consistent with a process that reduces variability by dividing longer intervals into shorter ones[42,43].

This observation suggests, notably, that the monkeys' behaviour might also exhibit subscalar variability. To test this prediction, we constructed two generative Bayesian models (Fig. 4e,f). Both models combined the prior distribution of vector lengths (Fig. 1b) with noisy measurement and used the posterior mean to generate Bayesian estimates (Extended Data Fig. 9). The two, however, made different assumptions about the form of timing variability during the mental navigation epoch. The first model assumed that the standard deviation scales with the mean, consistent with ignoring landmarks (no reset). The second model assumed that longer temporal distances are divided into multiples of 0.65 s, which leads to a sublinear relationship between the standard deviation and the mean (see Methods), consistent with having intermediate resets associated with endogenous landmarks. Fitting these two models to the monkeys' behaviour, we found that the model with intermediate resets was significantly better than the one without resets at capturing behaviour (Fig. 4g). This finding provides further evidence that neural mechanisms that support timing through mental navigation are qualitatively different from those associated with classical interval timing. Together, these results provide a mechanistic understanding of how mental navigation using endogenous landmarks introduces resets into neural dynamics, and how the resulting dynamics reduce behavioural variability.

Our results provide compelling evidence for the recruitment of a cognitive map in the EC during mental navigation, consistent with previous findings from imaging experiments in humans[25,26]. The closest result to our work comes from a virtual reality experiment in humans, which found a group of 'memory-trace' cells in the EC that were activated shortly before participants pressed the button in anticipation of reaching an invisible landmark[23]. Because the target landmark in that study was invisible, it is possible that the reported memory-trace cells overlap with the periodic neurons in our study. However, cells that are activated before a motor response are widespread in the neocortex and might be qualitatively different from those that support endogenous activity generation in the absence of a motor response. For example, the same study found evidence of similar memory-trace cells in the cingulate cortex, and previous work has found such activations in other brain areas, including the parietal cortex and the medial frontal cortex[44]. Indeed, we found a similar motor preparation signal in our dataset near the end of the trial at the time of joystick offset, regardless of whether neurons had periodic activity or not (Fig. 2 and Extended Data Fig. 10). Therefore, further work is needed to test the relationship between the cells in the EC that are activated shortly before a motor response and those that drive endogenous periodic activity.

Our work raises several questions about the architecture and function of the EC. The periodic activity and preserved cell–cell relationships in the EC are reminiscent of GCs in rodents[12,13,45], and implicate functionally homologous low-dimensional continuous attractor architectures, even though they might be a functionally distinct ensemble. Alternatively, the periodic neurons could be of the same class of neurons recently discovered in the rodent MEC[46] and the human EC[47], which exhibit periodicity ranging from seconds to minutes. However, without rigorous behaviourally controlled studies, it is difficult to ascertain whether these newly found periodic neurons are all the same cell type or whether they all serve the same functional purpose. The endogenous nature of the periodic dynamics indicates that these neurons receive an endogenous velocity input. This input can be readily supplied by a corollary discharge or reafference of the motor command for the continued deflection of the joystick. However, for the periodic activity to

match the temporal organization of memorized landmarks, the system must calibrate the velocity gain or adjust the attractor landscape within the EC, or both, so that the input leads to the appropriate periodicity. We verified the plausibility of this scheme with Hebbian learning, but other synaptic plasticity mechanisms found in this system are potential alternatives[48].

Many EC neurons exhibited ramping activity[49]. This ramping activity may be generated within the medial temporal lobe or be supplied externally. Within the medial temporal lobe, recurrent interactions among neurons with heterogeneous tuning in the EC[9,19,50–54] and hippocampus[20,21] could lead to such ramping activity. For example, adjustments in the overall inhibitory tone can create a ramping activity in the EC path integration network. Alternatively, the ramping activity might be supplied by other cortical circuits in the parietal and frontal areas that are known to generate such activity during timing tasks[32,44,55,56]. Interactions between continuous ramping and discrete periodic activity within the EC might be important for calibration and learning[57]. The fact that the phase and the periodicity of the endogenous activity in the EC were predictive of the monkeys' trial-by-trial timing behaviour is consistent with such ongoing calibration.

Our results regarding distance coding near joystick onset provide support for the possibility that the brain uses the presented start and target landmarks during the inference epoch to determine the direction and distance to the target. Distance coding near joystick offset, in turn, carries information about the number of landmarks traversed. An alternative possibility could have been that the EC encodes individual images along the landmark line. If so, we would expect firing rates at the time of joystick offset to carry information about the target image instead of the distance travelled. The key difference between these two computational strategies is that the latter is highly contextualized and would only work for a specific image set, whereas the former would facilitate generalization to new image sets so long as the image sequence is structured in the same way. The offset coding scheme in the EC is consistent with the idea that the monkeys learned the general structure of the task (Extended Data Fig. 4f,g). On the basis of previous work comparing activity patterns in the EC and the hippocampus[58–60], it is plausible that the image coding scheme might be present more strongly in the hippocampus.

Our modelling work also makes predictions for future experiments. First, the presence of plastic synapses between the EC and putative 'landmark' cells is an important assumption in recent models of the medial temporal lobe[27,35–37,40]. Our work highlights a potential role for this plasticity in mental navigation and underlines the need to investigate the biological basis of this mechanism. Second, our model remains agnostic as to whether the system can flexibly integrate different velocity inputs (for example, different joystick speeds) by adjusting the velocity gain or whether it would need considerable additional learning by adjusting the attractor landscape. This question can be readily answered by varying the speed on a trial-by-trial basis and providing visual feedback about the speed (for example, by a flow field) without revealing the intervening landmarks. Third, EC neurons exhibited a periodic Fano factor. We were able to account for this phenomenon in our model on the basis of the reset-like effect of endogenous landmark input on network dynamics. This observation indicates that landmark-dependent error-reducing mechanisms that have long been noted in spatial navigation[38] could apply to cognitive computations as well, and thus deserve further investigation.

Finally, we consider the implications of our findings for other behavioural contexts and neural systems. Many behaviours, such as silent counting and mental rehearsal, involve traversing through structured memories without sensory input. Although we know nearly nothing about the precise neural mechanisms of these high-level cognitive behaviours, we note that they have computationally analogous components to our mental navigation task. For example, silent counting might rely on dynamics similar to what we have discovered in the EC:

a timer with intermittent resets. If so, the behaviour might rely on a continuous attractor neural system that treats silent counts as abstract landmarks. This would also explain why counting reduces variability[42]. With these considerations in mind, we hope that our work will contribute to a circuit-level understanding of cognitive processes within the memory system.

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

## Methods

### Monkeys

All experimental procedures conformed to the guidelines of the National Institutes of Health and were approved by the Committee of Animal Care at the Massachusetts Institute of Technology. Experiments involved two male, awake, behaving monkeys (species: *Macaca mulatta*; ID: A and M; weight: 8.4 kg and 11.5 kg; age: 6 and 11 years old). The monkeys were head-restrained and seated comfortably in a dark and quiet room, and viewed stimuli on a 58.4-cm monitor (refresh rate: 60 Hz). Eye movements were registered by an infrared camera and sampled at 1 kHz (Eyelink 1000, SR Research). Hand movements were registered by a custom single-axis potentiometer-controlled joystick, the voltage output of which was sampled at 1 kHz (PCIe6251 board, National Instruments). The MWorks software package (https://mworks. github.io/) was used to present stimuli and to register hand and eye position. We used 32- and 64-channel laminar probes (V-probe, Plexon) for neurophysiology recordings driven by a motorized micromanipulator (Narasighe) through a biocompatible cranial implant. Analysis of both behavioural and spiking data was performed using custom MATLAB code (Mathworks).

### Tasks

**NTS.** Each trial begins with the monkey fixating a central spot of size 0.5 degrees of visual angle (dva). Next, an image sequence is presented above the fixation point. The sequence consists of six equidistant landmark images (inter-landmark distance of 6 dva), denoted $I_1$ to $I_6$. The sequence is initially shifted horizontally such that a randomly chosen image ($I_i$) from the sequence appears directly above the fixation point. We refer to this landmark as the start landmark. Next, we present a different randomly selected image ($I_j$) from the sequence directly below the fixation point, which we refer to as the target landmark. The start and target landmarks stay on the screen for a variable time (400–1,400 ms, uniform hazard). Afterwards, a change in the colour of the fixation point serves as the Go cue, instructing the monkeys to deflect a 1D joystick that moves the entire image sequence above the fixation point leftward or rightward at a constant speed (10 dva s$^{-1}$). The monkey must bring the joystick to its central position and stop the movement when the image above the fixation point matches the target landmark image below the fixation point (see Extended Data Fig. 1a and Supplementary Video 1). One way to think about the task is that the monkey controls the environment through joystick movement—rather than controlling their own movements through the environment. This design feature enabled us to extend the image sequence beyond the limits of the display. Throughout the trial, only images that were within the width of the monitor were visible. Trials are separated by an ITI (500–1,000 ms, uniform hazard). Because each trial started only after the monkeys acquired fixation for 200 ms, we were able to grab several ITI segments longer than 4 s. We pooled these segments of data for comparative analyses (for example, Extended Data Figs. 2b and 3e). In essence, the monkeys must produce a 1D vector $\mathbf{v}_p$ that matches the vector extending from the start landmark to the target landmark, denoted $\mathbf{v}_a$. Because the movement speed is constant, these vectors can be expressed as signed numbers whose magnitude corresponds to the temporal distance between images and whose sign represents direction. We designated rightward- and leftward-pointing vectors as positive and negative, respectively.

**MNAV.** We used NTS to help the monkeys learn the basic task contingencies, inter-landmark distance, image sequence and joystick speed. Next, we started the training on the main MNAV task, which is identical to NTS except for the following two crucial differences. First, during the initial presentation of landmarks, all of the landmarks in the sequence except the one right above the fixation are invisible. Second, between joystick onset and joystick offset, all landmarks, including the one above the fixation, are made invisible. After the joystick offset, the landmark closest to the fixation point is presented (Fig. 1a and Supplementary Video 2). Monkeys receive a reward if the relative error defined as $|v_p - v_a|/v_a$ is smaller than a criterion value of $0.08 \times |v_a|$. If not, the monkeys are given a second and last chance to produce a corrective vector. The second attempt furnishes one-quarter of the original reward if the relative error is smaller than the criterion. When rewarded, reward decreased linearly with relative error. When the monkeys aborted the trial by deflecting the joystick before the go cue, a time out of 5 s was added to the ITI. This was done to discourage the monkeys from purposely aborting long-vector trials. Throughout the paper, only single-attempt trials are considered correct trials. In Extended Data Figs. 2c and 5a, we considered incorrect trials, which are defined as trials in which monkeys made more than one attempt regardless of whether the monkey completed the trial in its second attempt. For the analysis in Extended Data Figs. 2c and 5a, we obtained the data from the first attempt of incorrect trials.

**Performance criterion.** Performance was quantified by the slope of the regression line (red) relating $v_p$ to $v_a$. We used the regression slope of 0.8 as a training criterion. After the monkeys reached this criterion in at least one session, we continued the training for one more week (seven sessions) to ensure stable behaviour (at least one other session in which the monkey reached the criterion) before testing the monkeys on generalization pairs.

### Electrophysiology and preprocessing

Monkeys were trained without a recording chamber. All recordings were performed after the completion of training and verification that monkeys could generalize. Afterwards, a recording chamber that provided access to the EC was implanted. We located the EC on the basis of stereotaxic coordinates and structural magnetic resonance imaging (MRI) scans acquired from both monkeys after the chamber implantation[61]. To target the EC reliably, we used a grid system inside the recording chamber. We registered grid holes relative to the brain using an MRI scan in which the holes were filled with an MRI contrast agent (5 mg ml$^{-1}$ gadolinium + 10 mg ml$^{-1}$ agar). We used the registered grid system together with readings of anatomical landmarks along the penetration path to target the EC accurately.

We recorded extracellular neural activity in the EC acutely across 32 sessions (A, 17; M, 15) using multi-channel linear V-probe array electrodes from Plexon. All channels had an impedance of 275 (±50) kΩ. The configurations of the electrode contacts are shown in Supplementary Table 3. Recorded signals were amplified, bandpass filtered, sampled at 30 kHz and saved using the OpenEphys data acquisition system (OpenEphys). Using Kilosort 2.0 software[62], we isolated 1,478 single units and multi-units (A, 614; M, 864). We used Kilosort software to detect and automatically sort spikes. We used a Python-based GUI (phy) to verify and sort the output of the Kilosort algorithm manually. We first looked for spike artefacts that appeared in all channels and discarded them. We then looked for spikes that were unstable during a certain duration within a session. If nearby channels had clusters of spikes during those durations, we merged the two clusters of spikes if (i) they had a high correlation of spike waveform template (Pearson's correlation > 0.9) and (ii) the principal components (PCs) of spike waveform features were visually overlapping. Next, if a given cluster of spikes clearly showed two sets of waveforms and the PC space also exhibited two clusters of cloud, we split the spikes by manually drawing a line on the PC space to separate the two clusters maximally. We included both single units and multi-units in our analyses. We considered multi-units as those clusters that had no more than two zero crossings.

### Analysis of neural data

We focused our neural data analysis on the subset of task-modulated neurons defined as those whose firing rate either exhibited periodicity

during navigation or was modulated by temporal distance during joystick onset or offset (see below for the corresponding analyses). To plot firing rates, we smoothed spike counts in 1-ms bins using a Gaussian kernel with a standard deviation of 100 ms. Because of variability in the produced vectors, trials associated with the same condition (that is, same direction and distance) had variable lengths. To compute trial-averaged firing rates for each condition, we used 40-ms bins for the median produced interval and appropriately stretched or compressed bins for shorter and longer trials[32].

## Periodicity and ramping

We computed a PI for every EC neuron using a similar procedure to that used for computing gridness scores during spatial navigation tasks[30,45]. (1) We pooled firing rates for trials requiring mental navigation over at least three landmarks to ensure that trials were long enough to compute periodicity. (2) We truncated trials at 500 ms before the joystick offset to ensure that our estimate of periodicity was not biased by the associated large anticipatory response (see Fig. 2a). (3) We detrended firing rates using linear regression fits to the firing rate profile so that ramping activity would not mask the presence of periodicity. (4) We computed an average ACG for each neuron by averaging the single-trial autocorrelation function of firing rates at lags between 0 ms and 2,400 ms. (5) To detect periodicity, we computed the correlation between the ACG and shifted ACG for varying lags ranging from 0 ms to 1,300 ms. (7) We defined the PI at each lag as the difference between the ACG for that lag and the ACG for half that lag. This procedure is analogous to how gridness scores are computed, with the difference that instead of a two-dimensional spatial ACG, a one-dimensional temporal ACG is used. To evaluate the significance of the PI for each neuron, we also created a null distribution for PI using surrogate data generated from a zero-mean Gaussian process (GP) with a squared exponential kernel (maximum variance 1; length constant 100 ms). To match the smoothness of the GP to our smoothed firing rate, the length constant parameter of the squared exponential kernel was equal to the width of the Gaussian smoothing kernel (100 ms) used for smoothing the firing rates of EC neurons. We then passed the GP surrogate data through a non-homogenous Poisson process and smoothed the resulting spike train to obtain our surrogate GP null data. We repeated this process 1,000 times to obtain a distribution. A neuron was classified as periodic if its PI at any lag was higher than 2 standard deviations from the PI obtained from surrogate data and if the total number of pooled trials was higher than 15. The trial count threshold was applied to remove spurious periodicity arising from low signal-to-noise firing rates. The results are robust to the choice of minimum trials. We verified that our results and conclusions were unaffected when the trial count threshold was raised (for example, to 35 or 75). For a neuron with significant PI, its periodicity was quantified as the lag at which the PI was the maximum.

Ramping activity was quantified by fitting linear regression to the average firing rate pooled across long-distance trials (more than two image–distance trials). A neuron was considered significantly ramping if the $F$-statistic of the linear model was significant ($P < 0.001$) compared to a constant model.

## Quantification of the phase of local bumps of activity

To find the phase of localized bumps of activity with respect to joystick offset, we detected the phase of local maximum within a one-second window before joystick offset. We bootstrapped over 100 repeats of subsampled trials and calculated the distribution of the mean phase. We created a null dataset by circularly shifting each trial's spike train by a random length and repeating the phase detection steps.

To find the phase of the second-last peak, we detected the phase of the local maximum in the one-second window before the first peak relative to the joystick offset. Similarly, the phase of the third-last peak was the phase of the local maximum in the one-second window before the second peak relative to the joystick offset.

We computed the mode of the absolute phase distribution for each periodic neuron whose phase distribution was different from its shuffled control (KS test, $P <<< 0.001$).

## Modulation of temporal distance

For each neuron, we quantified the degree to which firing rates at joystick onset and offset were modulated by the produced temporal distance. For each neuron, we sorted the trials into 15 bins according to the produced vector length ($\mathbf{v}_p$), and computed the regression coefficient relating those lengths to the corresponding firing rates, both at joystick onset and offset. A neuron was considered to exhibit significant temporal distance modulation if the regression slope at either onset or offset was significant ($F$-statistic at 95% confidence).

We used a four-way ANOVA to quantify the effect of temporal distance, start landmark, target landmark and direction on average firing rate over a window of −500 ms to −300 ms relative to joystick offset. To compare the effect of distance and start (or target) landmark, we used a standard $t$-test on the distribution of the difference of $F$-statistic for distance factor and for start (target) landmark factor across all of the neurons with a significant effect of either distance or start (target) landmark. In one monkey, neural data were collected when the monkeys performed mental navigation over two different sequences. We verified that the results of the ANOVA analysis comparing distance coding to image coding were independent of whether the data were segregated or combined across the two sequences. We subsequently also verified that the findings of periodicity and onset–offset coding were independent of whether the data were segregated or combined across the two sequences.

The same four-way ANOVA model was also applied to the slope of the firing rate. Firing rates were extracted over a window of −550 ms to −250 ms relative to joystick offset. Slopes were estimated by fitting a line to the firing rate within that window. We compared the strength of distance encoding by mean firing rate to that by the slope of firing rate by comparing the distribution of $F$-statistics obtained from the two ANOVA models—one with firing rate and the other with the slope of firing rate as the dependent variable.

## Targeted dimensionality reduction

We used a regression analysis across the population of neurons to identify the dimension in which firing rates encode temporal distance[63] in two time windows: 200 ms before joystick onset (−200 ms to 0 ms) and 500 ms to 300 ms before joystick offset (−500 ms to −300 ms). We first centred the responses of each neuron by subtracting its mean response across the two windows of interest before joystick onset and offset. We then computed the regression line relating the firing rates to the temporal distance:

$$r_{it}(k) = \beta_{it}(k)\mathrm{dist}(k)$$

Here, dist($k$) is the ordinal distance on trial $k$ (dist: 1 to 5) and $r_{it}(k)$ is the centred firing rate for trial $k$ at time $t$ and for neuron $i$. For $N$ neurons, we had a $N \times t$ matrix of regression coefficients. We took the norm of this matrix along the dimension of time to find the time point at which the coefficients were maximum. The vector of regression coefficients is considered the optimal axis for coding ordinal distance. Next, we projected the matrix of held-out trials for the same neurons onto the optimal distance axis (Fig. 3c,d).

We quantified the linear encoding of distance by measuring the variance accounted for by a linear model ($R^2$) with a dummy independent variable, dist = [1,2,3,4,5]. To compare the encoding of the ordinal distance across the two epochs—joystick onset and joystick offset—we created a bootstrapped distribution (50 resamples) of distance projections averaged over the 200-ms windows.

To control for the effect of the shortest distance inflating the linear readout of distance coding, we performed the temporal distance

coding analysis (single neurons, Fig. 3b and population analysis, Fig. 3d) excluding the distance of 1. In both cases, we found significant linear coding of distance near the joystick offset.

## Cross-validated $R^2$

We calculated the cross-validated $R^2$ metric for single neurons in the following steps. (1) Divide all trials randomly into training ($y_{train}$) and test trials ($y_{test}$) with mean firing rate data over a 200-ms window before joystick offset or onset. (2) Estimate a linear regression slope for training data (dummy variable: $X = [1 2 3 4 5]$ for the five distances). (3) Calculate the predicted training data from the regression slope: $y_{pred} = \beta X$. (4) Calculate the cross-validated $R^2$ as variance accounted for by test data $y_{test}$:

$$R^2 = 1 - \frac{\Sigma(y_{test} - y_{pred})^2}{\Sigma(y_{test} - \overline{y_{test}})^2}$$

## CAN model

We tackled the circuit-level modelling of our task in two steps. First, we examined the conditions under which GC activity could serve as endogenous landmarks. To do so, we constructed a model with multiple GC modules with different periodicities and phases and a hypothetical LM. The LM received both external input ($I_{ext}$) from visual stimuli and internal input ($I_{int}$) from GCs. The synaptic weights from GC to LM were subjected to plasticity. Learning proceeded in two stages, first with both $I_{ext}$ and $I_{int}$, mimicking conditions in NTS, and later with $I_{int}$ only, mimicking conditions in MNAV. At first, synaptic weights were sampled randomly from a normal distribution. Throughout learning, synapses were updated at every time step using Oja's rule. To ensure learning was stable, we (1) used a sufficiently small learning rate (that is, smaller than $1 \times 10^{-7}$) and (2) normalized the weights for each module such that they were always centred at zero. This learning scheme selectively strengthened inputs from the subset of GCs whose periodicity and phase match external input (Fig. 4b and Extended Data Fig. 6b,c).

Having established that learned $I_{int}$ could emulate $I_{ext}$, we next constructed a CAN model of GCs adapted from a previous study[14] (code at: https://fietelab.mit.edu/code/) to compare the effect of attractor dynamics in path integration versus mental navigation. The model GCs have difference-of-Gaussian connectivity kernels, with centres shifted clockwise or counterclockwise, and perform path integration when driven by matching velocity inputs (for example, left-shifted cells receive leftward velocity inputs). We first characterized the path integration behaviour of this model in the presence of variable velocity input. To do so, we simulated the network when the velocity input on each trial was perturbed by Gaussian noise and quantified the time it takes for the network to reach different distances measured in terms of network phase (Fig. 4c, black).

Next, we constructed a new CAN model to capture mental navigation behaviour by providing the additional $I_{int}$ from the subset of GCs whose periodicity and phase match $I_{ext}$ (Fig. 4c, red). Because the landmark input ($I_{int}$) transiently stops the velocity drive of the network, we scaled the velocity input such that the average time taken to reach a desired state is the same for the two models. To do so, we first did a grid search to identify the optimal velocity at which the learned landmark periodicity matched our experimental periodicity of 0.65 s (Extended Data Fig. 6d). This step also verified our models' validity at a range of input velocities. Under the optimal velocity, we found the distance traversed (in terms of phase state) by the network with landmarks in 0.65 s and its multiples. We then did a second grid search in the model without landmarks to identify appropriate velocity inputs such that, on average, the model takes the same duration to reach the corresponding distances. We performed the second grid search five times independently for the five temporal distances

(that is, multiples of 0.65 s). Finally, at each of these base intervals, we compared the overall variability of the two models (Fig. 4d) while holding the average temporal distance equal (Extended Data Fig. 6e). We repeated the entire procedure of comparing variability across three different Weber fractions to ensure that the result was robust to networks instantiated with different noise levels (Extended Data Fig. 6e).

## Bayesian model of behaviour

We fit two Bayesian observer models to behavioural data (Extended Data Fig. 9a). Both models combine the likelihood function associated with a noisy measurement with the experimentally imposed prior distribution and use the posterior mean, $t_e$ (that is, Bayes least squares estimation) to estimate the temporal distance[64]. Both models also assume that the navigation epoch introduces additional variability. The variability in the path integration model is assumed to follow a scalar property[41,42] (Extended Data Fig. 9a) as follows:

$$\sigma = w_p t_e$$

By contrast, the mental navigation model divides longer temporal intervals into multiples of the base interval ($t_o$), which in our experiment is 0.65 s. In this case, by variance sum law, the total variance $\sigma^2$ is

$$\sigma^2 = (w_p t_o)^2 D = (w_p^2 t_o)(t_o D)$$

where $D$ is the inter-landmark temporal distance.

Accordingly, the standard deviation grows as the square root of the total interval:

$$\sigma = (w_p \sqrt{t_o}) \sqrt{(t_o D)} = w_{pc} \sqrt{t_e}$$

where $w_{pc}$ is defined as $w_p \sqrt{t_o}$.

We used surrogate data generated by each model to verify that our fitting procedure could correctly identify the generative model (Extended Data Fig. 9b,c). Next, we compare the model fits to behaviour. The models used to fit the behaviour were augmented to include an offset term, $b$, to account for the overall bias in produced temporal intervals. We used maximum likelihood estimation to fit the models to behaviour on each session and used the predicted bias and variance to compare the models.

For behavioural modelling with Bayesian least squares optimization, we first used a probabilistic mixture model to exclude outliers. The model assumed that each $v_p$ was either a sample from a task-relevant Gaussian distribution or from a lapse distribution, which we modelled as a uniform distribution extending from 0 to $3v_p$. Using this model, we excluded any trial in which $v_p$ was more likely to be sampled from the lapse distribution. The outlier-free data then went into the modelling algorithm.

The plots in Fig. 1 and Extended Data Fig. 1 and the calculation of regression slopes in Fig. 1b (example sessions), Fig. 1c (distribution of slopes) and Fig. 1e (generalization plots) include all trials, with no trial exclusion applied.

## Fano factor of CAN model units

The model's Fano factor was computed by first simulating it 50 times under noisy velocity input and calculating the mean and variance of each unit activity across the 50 simulations. We then followed the autocorrelation procedure described above for the PI to compute the periodicity of both the Fano factor and the activation time series of model units.

## Reporting summary

Further information on research design is available in the Nature Portfolio Reporting Summary linked to this article.

## Data availability

The data used to generate the figures are available on DANDI at https://dandiarchive.org/dandiset/000897. Source data are provided with this paper.

## Code availability

The code used for analyses is available at https://github.com/jazlab/Neupane_Fiete_Jazayeri_mental_navigation. This page also contains the link to raw datasets uploaded to DANDI.

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

**Acknowledgements** S.N. is supported by NSERC PDF-516867-2018, FRQNT B3X-258512-2018; I.F. is supported by the National Institutes of Health (NIH) (NIMH-MH129046); and M.J. is supported by the NIH (NIMH-MH129046), the Paul and Lilah Newton Brain Science Award and the McGovern Institute. We thank A. Kinkabhwala for sharing with us a previously published dataset[28] and the associated analysis code.

**Author contributions** S.N. and M.J. conceived the study. S.N. performed all experiments and data analyses. S.N. and M.J. developed the models and performed model simulations. I.F. and M.J. oversaw the analyses. S.N., I.F. and M.J. wrote the manuscript. M.J. supervised the project.

**Competing interests** The authors declare no competing interests.

**Additional information**
**Correspondence and requests for materials** should be addressed to Mehrdad Jazayeri.

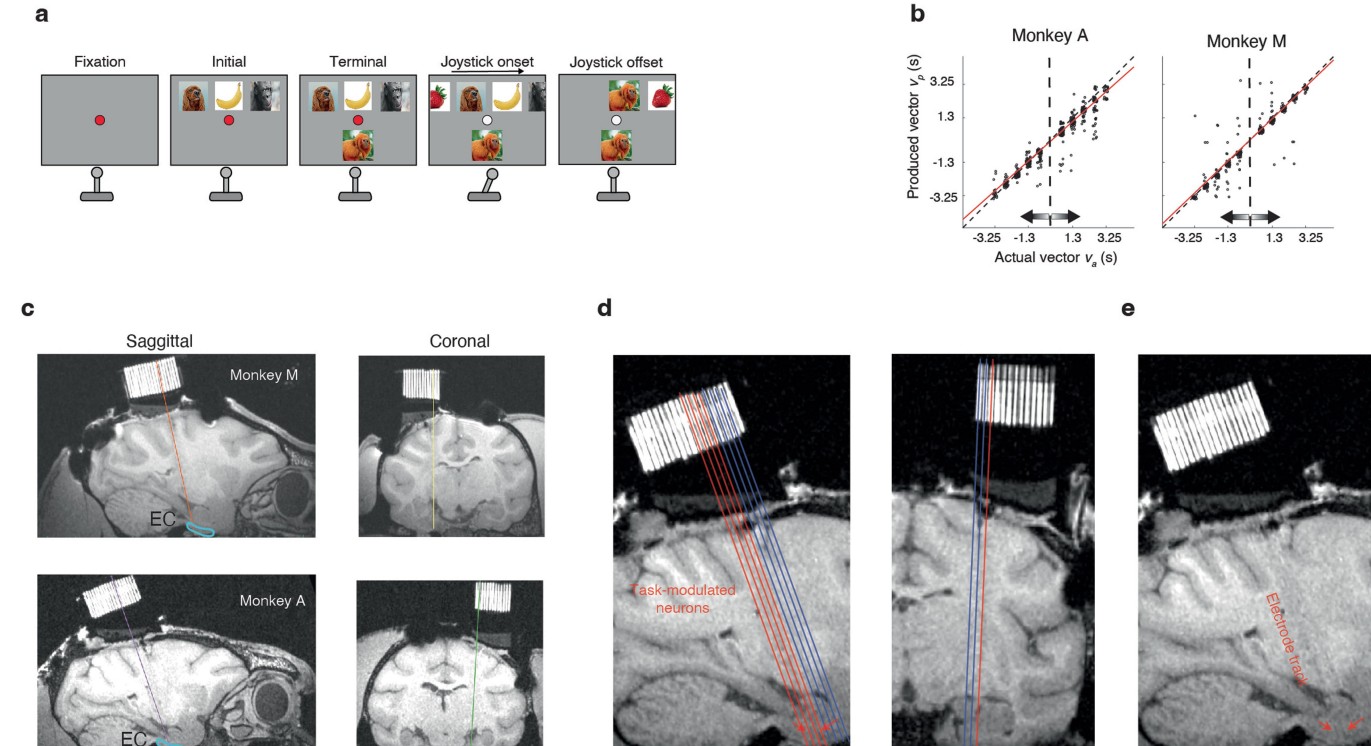

**Extended Data Fig. 1 | NTS task behaviour and recording site. a**, Sequence of events during a trial of the NTS task. The sequence is identical to the MNAV task (Fig. 1a), except that during joystick deflection, all images and their movement are visible. **b**, Behaviour in a representative session quantified in the same manner as the MNAV task (Fig. 1b). **c**, Structural scan of a monkey M (top) and monkey A (bottom) together with the recording grid across sagittal and coronal anatomical planes. **d**, Tracks through the grid (red: tracks with task-modulated neurons). Anatomical coordinates: 9.5–10.5 mm from the midline, 14–16 mm anterior to ear bar zero (EBZ) and centred on EBZ along the vertical axis. **e**, Post-recording scan showing electrode tracks and recording sites targeting periodic neurons in the EC.

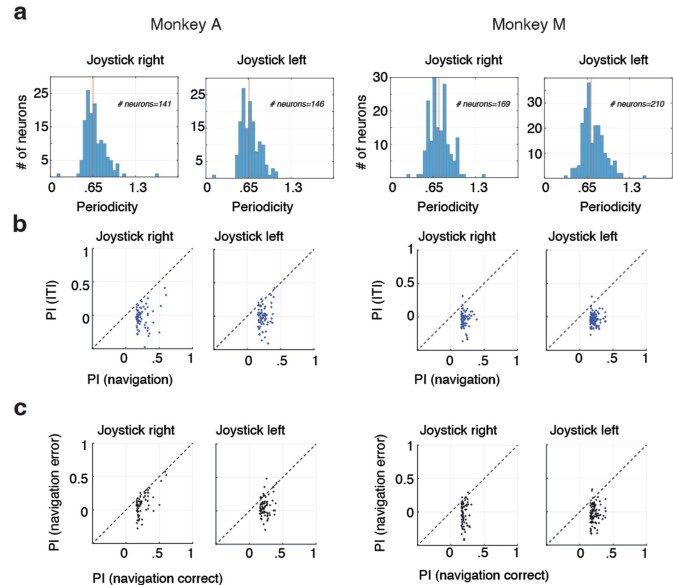

**Extended Data Fig. 2 | Periodicity across sessions for different directions of navigation, epochs and trial types. a,** Distribution of periodicity of all periodic neurons for two monkeys and for two directions of navigation. **b,** Comparison of PI at 0.65 s during the ITI versus the navigation epoch. One-tailed paired *t*-test, monkey A, right: t(67) = 14.07, p<<.0001; A, left: t(75) = 14.02, p<<.0001; monkey M, right: t(73) = 16.93, p<<.0001; monkey M, left: t(103) = 26.56, p<<.0001. **c,** Comparison of PI at 0.65 s during error versus successful trials in the navigation epoch. One-tailed paired *t*-test, monkey A, right: t(67) = 10.58, p<<.0001; A, left: t(75) = 11.02, p<<.0001; monkey M, right: t(65) = 10.11, p<<.0001; monkey M, left: t(102) = 17.73, p<<.0001.

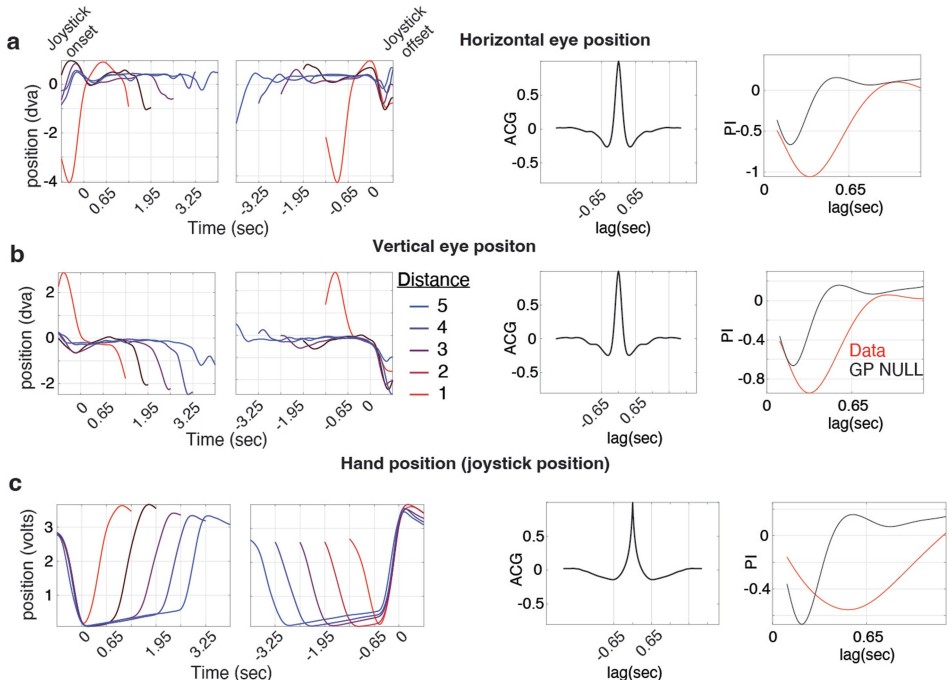

**Extended Data Fig. 3 | Eye and hand position time-course dynamics during the navigation epoch. a**, Left, time course of eye position between joystick onset and offset averaged across trials for horizontal eye position disaggregated for five different temporal distances (shades of red to blue). Middle, average ACG of the horizontal eye position time course. Right, PIs at different time lags for horizontal eye position together with the significance threshold (black) at two standard deviations above chance-level PI computed on Gaussian process surrogate data (n = 100 bootstraps, average PI at 650 ms = −0.25 is smaller than chance (one-tailed *t*-test(99) = −67.8, p<<.0001). **b**, Same as **a** for vertical eye position (n = 100 bootstraps, average PI at 650 ms = −0.44 is smaller than chance (one-tailed *t*-test(99) = −81.6, p<<.0001). **c**, Same as **a** for hand position measured with the voltage readout of joystick position (n = 100 bootstraps, average PI at 650 ms = −0.53 is smaller than chance (one-tailed *t*-test(99) = −138.1, p<<.0001).

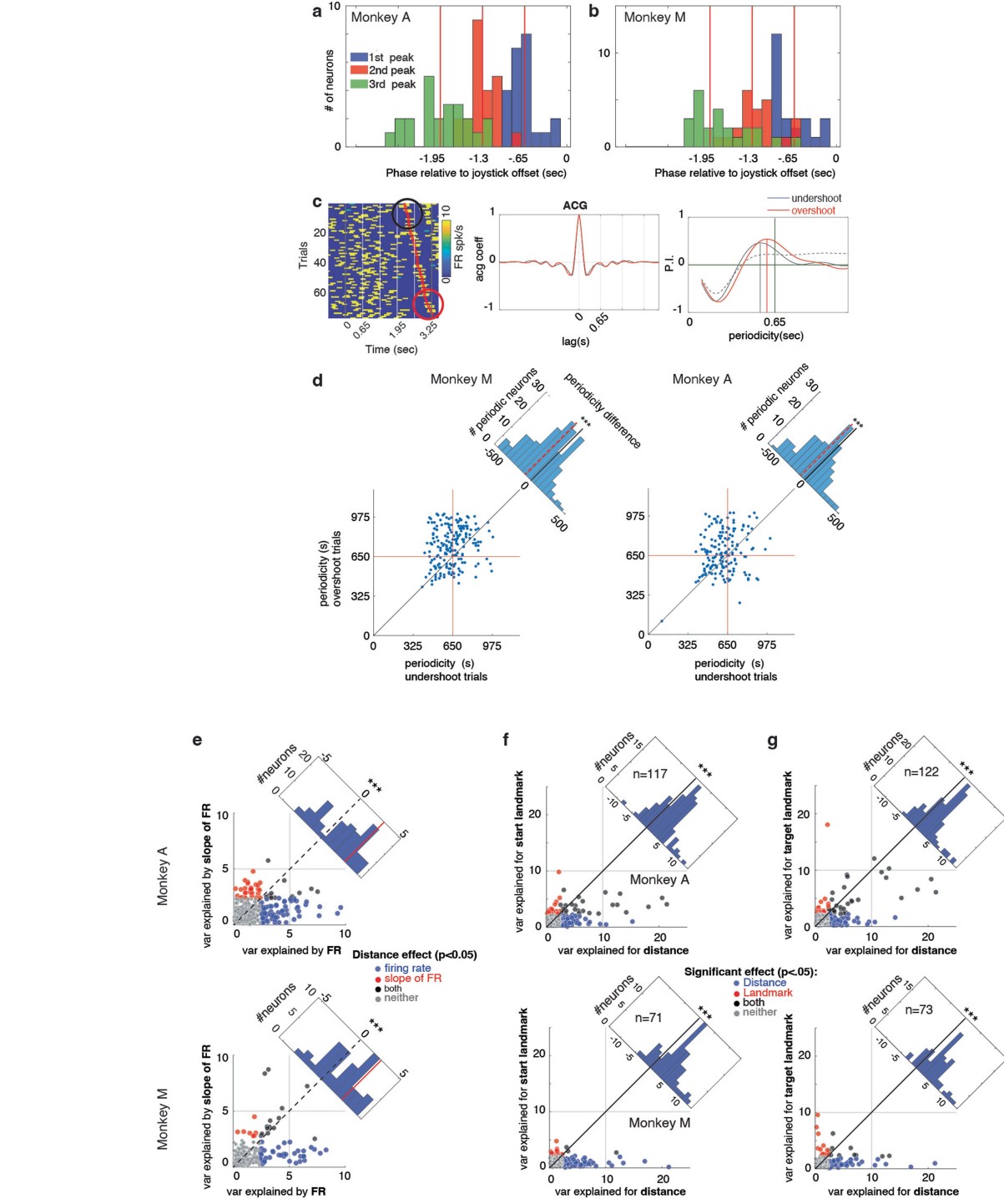

**Extended Data Fig. 4** | See next page for caption.

**Extended Data Fig. 4 | Relationship between monkeys' timing error and periodicity or phase, and encoding properties of EC neurons at joystick offset. a,b,** Distribution of the first (blue), second (red), and third (green) peak phase before joystick offset across neurons for monkey A (**a**) and monkey M (**b**). Same as Fig. 2f for the two monkeys separately. **c,** Left, raster plot of an example neuron with trials sorted according to the produced vector. The top (black circle) and bottom (red circle) correspond to undershoot and overshoot errors, respectively. Middle, ACG of undershoot and overshoot trials. Right, monkey A: PI at various lags for undershoot and overshoot trials shows higher periodicity for overshoot trials. **d,** To quantify the effect of periodicity on timing error, we focused our analysis on trials with a distance of 3, which were long enough to quantify periodicity (i.e. two full periods) and afforded enough repeats to gain statistical power. For these trials, we sorted the distribution of timing error in ascending order and pooled trials from the top and the bottom tertiles representing undershoot and overshoot trials. We then computed the periodicity for the two groups of trials following the same ACG procedure as in Fig. 2. Note that the number of neurons is smaller than the total number of periodic neurons because neurons with fewer than 15 trials in each bin were dropped from the analysis. **d,** Left, scatter plot of all periodic neurons' periodicity for undershoot trials against those for overshoot trials. Inset, distribution of periodicity difference. Right, same as left for monkey A. Across neurons, the response period was significantly larger for overshoot trials in both monkeys (rank-sum test; Z(194) = −5.80, p<<<.0001 for monkey M and Z(161) = −3.3,

p<<<.0001 for monkey A). **e–g,** We used ANOVA to analyse the degree to which different factors (distance, start landmark, target landmark, and direction) explain the variance of individual neuron's firing rates or the rate of change of firing rate (slope) prior to the joystick offset. Firing rates were extracted within a window of 200 ms before the joystick offset. Slopes were estimated by fitting a line to the firing rate within a window of 300 ms before the joystick offset. The two rows correspond to the two monkeys. For all plots, the metric of variance explained is the F-ratio for the corresponding independent variable in the ANOVA model. **e,** Variance explained by the firing rate plotted against variance explained by the slope of firing rate (blue: significant distance effect for firing rate, red: significant distance effect for slope, black: both, grey: none). Inset: distribution of the difference of variance explained showing that distance effect due to mean firing rate is significantly larger than that due to the slope of firing rate (two-sample $t$-test, p<<.0001). **f,** Variance explained for start landmark factor plotted against variance explained for distance factor (blue: significant effect for distance, red: significant effect for start landmark, black: both, grey: none). Inset: distribution of the difference of variance explained, showing that the distance effect is significantly larger than the start landmark effect (two-sample $t$-test, p<<.0001). **g,** Same as b for target landmark factor vs. distance factor (blue: significant effect for distance, red: significant effect for target landmark, black: both, grey: none). Inset: distribution of the difference of variance explained, showing that the distance effect is significantly larger than the target landmark effect (two-sample $t$-test, p<<.0001).

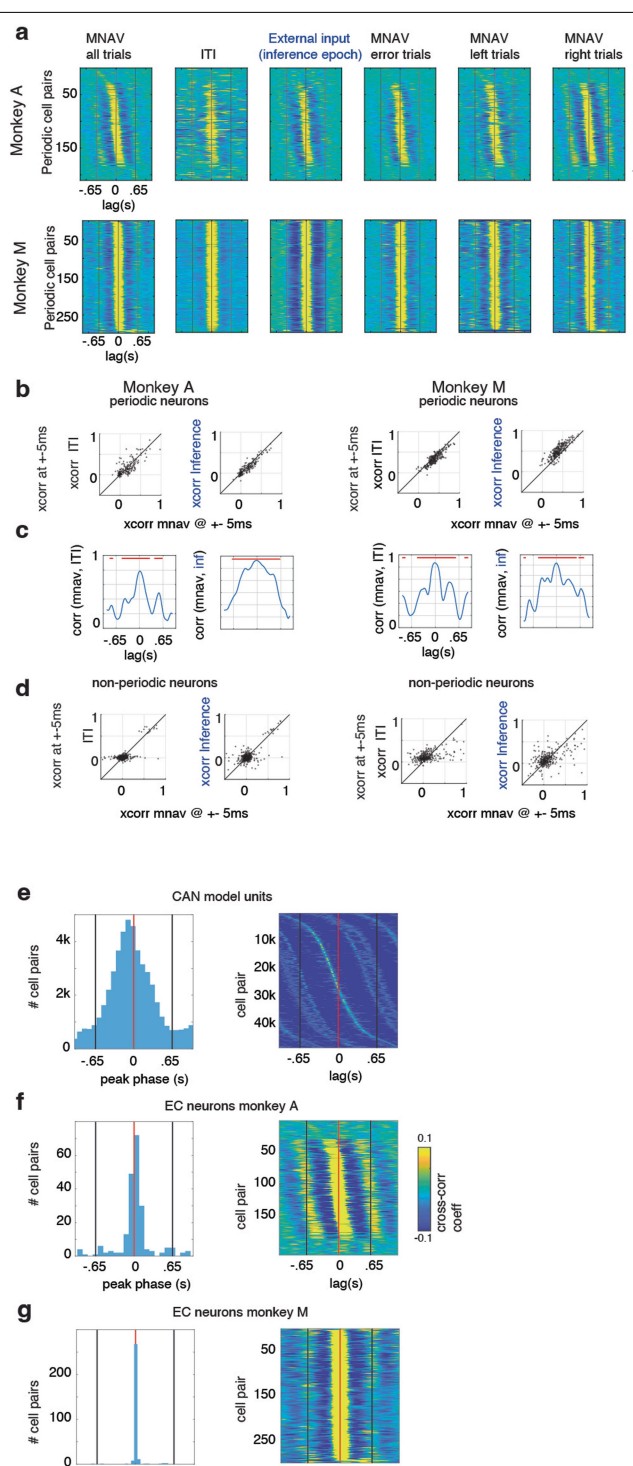

**Extended Data Fig. 5 | Cross-correlation structure of periodic neurons across epochs. a**, Cross-correlation structure of cell pairs for the top 25 periodic neurons in the window during mental navigation (first panel), during the ITI (second panel), during the presentation of start and target landmarks (third panel), during MNAV error trials (fourth panel) and during MNAV left and right direction trials (two rightmost columns), all sorted according to the peak correlation lag in the mental navigation window. Top: monkey A and bottom: monkey M. **b**, Cross-correlation values averaged over the lag of −5 to 5 ms for data from mental navigation plotted against data from ITI (left) and from the image presentation window (right). Left panels: monkey A, right panels: monkey M. **c**, Correlation between cross-correlation values during mental navigation epoch vs ITI (left) and vs inference epoch (right) at various lags (left). Left panels: monkey A, right panels: monkey M. Red stars denote the significance of Pearson's correlation at p < .05 with Bonferroni correction. **d**, Same as **b** but for sessions in which the neurons were non-periodic. **e**–**g**, Cell–cell cross-correlation analysis compared across model and neural data. Left, distribution of lags at which each cell pair's cross-correlation peaked for model (**e**), monkey A (**f**) and monkey M (**g**). Right, cross-correlation structure of simultaneously recorded periodic neurons, rank ordered based on activity during the navigation epoch for model (**e**), monkey A (**f**) and monkey M (**g**).

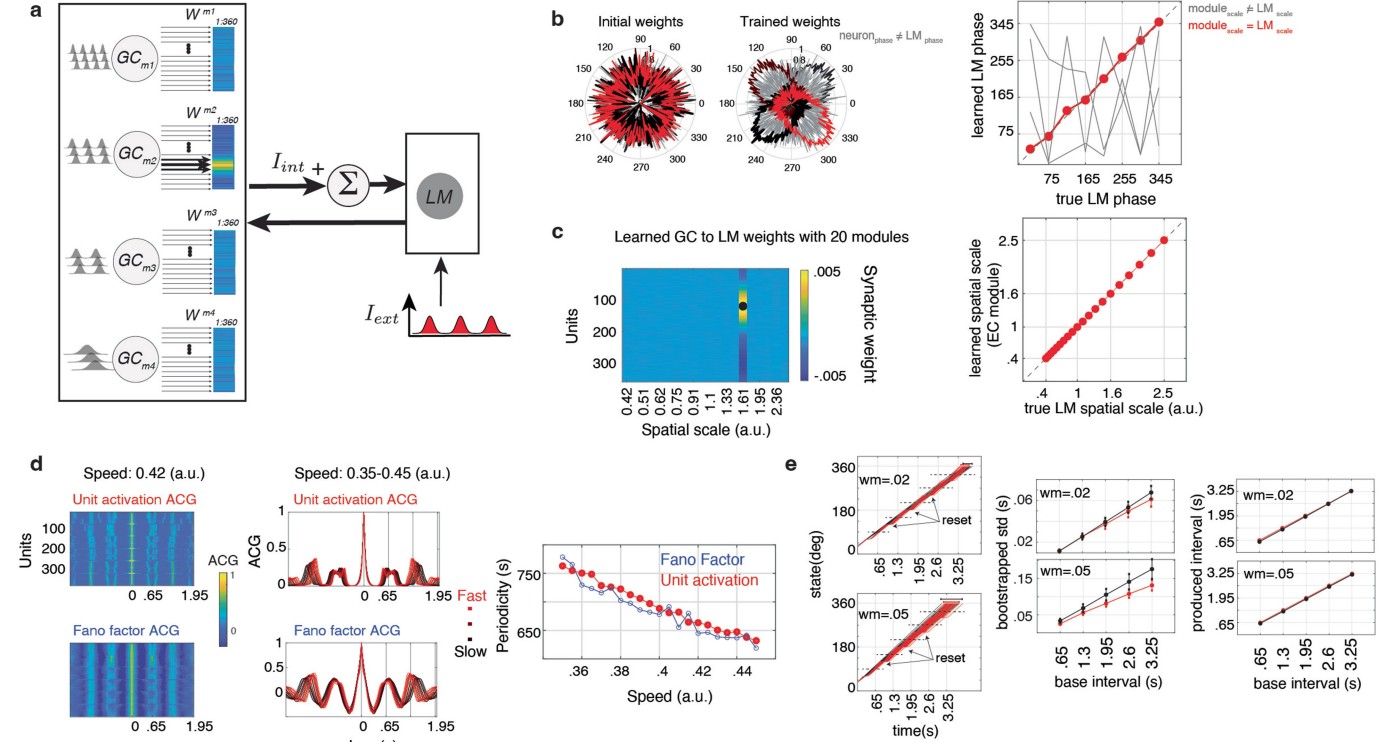

**Extended Data Fig. 6 | Model identifiability for the CAN model. a**, Schematics of the model, similar to Fig. 4a. GC modules receive input from a LM that receives both internal input ($I_{int}$) from GCs and external input ($I_{ext}$) from visual stimuli. The synaptic weights from GC to LM undergo Hebbian plasticity ('+') and undergo learning both in the presence of $I_{ext}$ mimicking conditions in NTS and in its absence mimicking conditions in MNAV. **b**, Learning of GC to LM connections in the presence of $I_{ext}$ with a specific periodicity and 4 different phases (i.e., 4 different model simulations). Left, synaptic weights from all neurons in all GC modules (m1, m2, m3, m4) are initially random. Middle, after learning, synaptic weights of those GC cells whose periodicity and phase match $I_{ext}$ (middle) strengthens. Right, tests of the model with landmark placed at 8 different phases. Learned landmark phase, computed as the phase of the GC neuron with maximum weight to the LM neuron, plotted as a function of the true landmark phase for the appropriate module (red) and all other modules (grey). **c**, Learning in a new instantiation of the model with 20 modules. Left, in the presence of $I_{ext}$ with a specific periodicity and phase (black circle), the model learns the correct phase and periodicity. Right, robustness of learning relative to landmark periodicity. The model learns to associate the module with the correct periodicity for a wide range of scales. **d**, Left, ACG of average unit activation (top) and ACG of Fano factor (bottom) of all units in the model under noisy velocity (average velocity: 42 a.u.). Middle, average ACG across all units for a range of velocity (red: fastest; black: slowest). Right, periodicity of unit activation and Fano factor both scale with the velocity input. **e**, Left, model simulations with (red) and without landmarks (black) under different levels of noise (Weber fraction, wm=0.2 and 0.5; wm=0.8 in Fig. 4c,d; avg. velocity: 42 a.u.). Dotted lines denote the distance traversed in 650 ms and its multiples. Arrows point to locations of endogenous resets in the network dynamics. Middle, bootstrapped standard deviation of temporal distance versus mean temporal distance at corresponding noise levels. Right, mean-matched temporal distances were achieved by adjusting velocity input to the models.

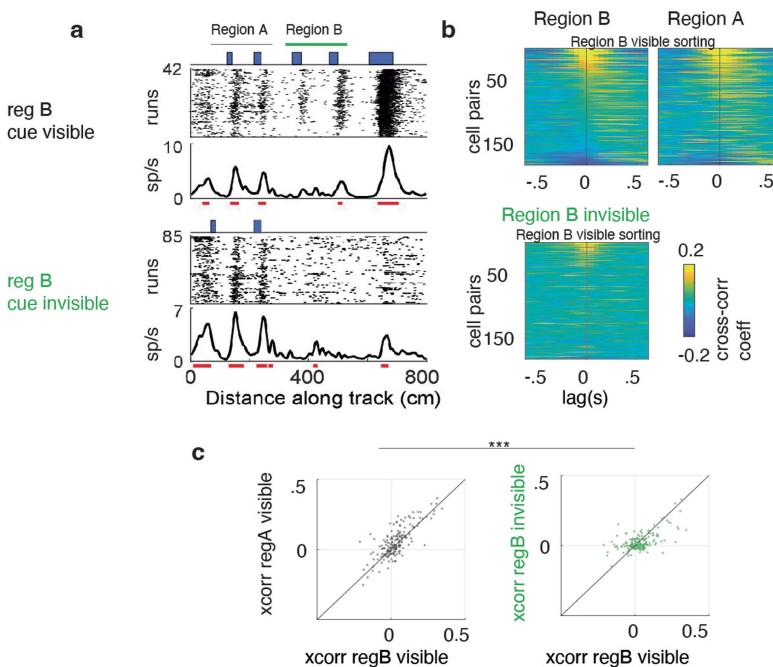

cross-correlation analysis on data from
Kinkhabwala et al (eLife 2020)

**Extended Data Fig. 7 | Cell–cell cross-correlation analysis on data from a previous study. a**, Reproduced Fig. 2a from Kinkhabwala et al.[28] showing the firing pattern of a landmark cell when landmarks were visible (top) and invisible (bottom). **b**, Cell–cell spike time cross-correlation across all the simultaneously recorded cell pairs for region B (left) and for region A (right), both sorted according to region B data. The bottom plot shows the same analysis performed on data for region B when landmarks were invisible. **c**, Cross-correlation values averaged over the lag of +−5 ms for data from region B when landmarks were visible and plotted against the values from region A (left) and region B with landmarks invisible (right).

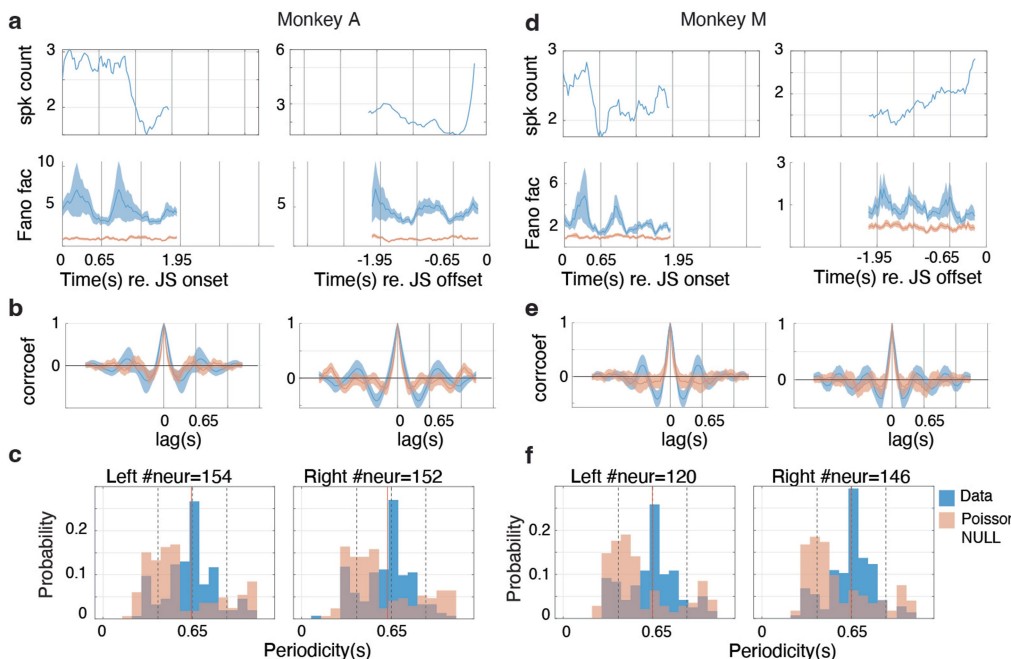

**Extended Data Fig. 8 | The Fano factor of EC neurons is periodic at the behaviourally relevant periodicity. a**, An example neuron (monkey A) showing a periodic Fano factor (bottom) but little or no periodicity in average spike count time series (top). **b**, ACG of Fano factor for the two example neurons (blue) and their corresponding (Poisson) null ACG (orange). **c**, Distribution of periodicity of the Fano factor for neurons with significant periodicity compared to their corresponding (Poisson) null data. Dotted lines denote the window within which significant difference was tested. We created a null distribution for each neuron by passing its mean firing rate through a Poisson process 100 times.

We calculated the distribution of the true Fano factor by bootstrapping 100 times. We then computed the autocorrelation of the Fano factor and the corresponding PI for both distributions. We then estimated the periodicity of each neuron that had a significantly higher PI in the window of 300 to 900 ms lag, compared to its corresponding *Poisson Null* (two-sample *t*-test with p < (.0001/20), with Bonferroni correction). The periodicities of the neurons with significant periodic Fano factor were tightly clustered at the behaviourally relevant periodicity of 650 ms. **d**–**f**, Same as **a**–**c** for monkey M. Note that plots **c**, right and **f**, right are also shown in Fig. 4h. They are included here for completeness and clarity.

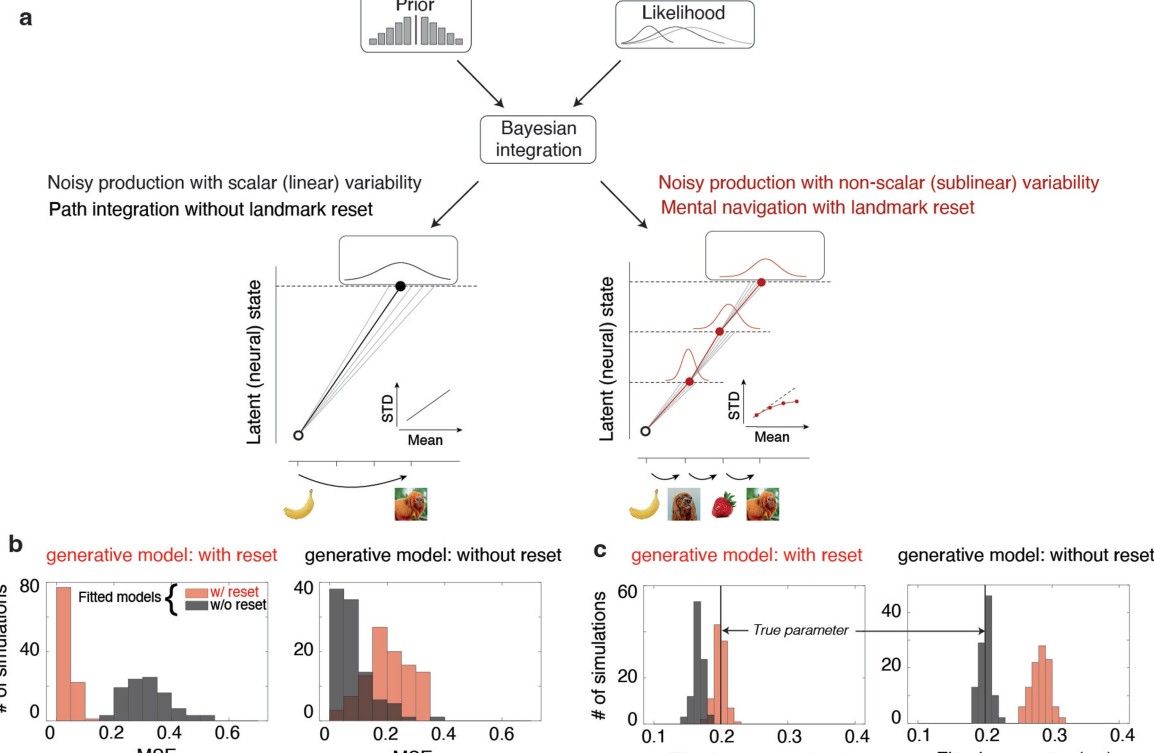

**Extended Data Fig. 9 | Description and identifiability of the two Bayesian observer models. a**, Schematic illustration of the Bayesian observer model that combines the prior (top left) with the likelihood function (top right: three example likelihood functions associated with three intervals) and uses the posterior mean to produce the desired interval. We considered two noise models for interval production. In one model (left), the standard deviation of noise scales with temporal distance. This model is consistent with path integration without incorporating landmarks resets. In the other model (right), the standard deviation increases sublinearly with temporal distance (variance increases linearly with temporal distance). This model is consistent with mental navigation with landmark resets. **b**, Distribution of mean squared error (MSE) between ground truth data generated from a chosen generative model and data generated from the two models fitted to the ground truth data. Left: ground truth data generated from a model with reset (blue). Right: ground truth data generated from a model without reset (red). **c**, Distribution of fitted parameter values (Weber fraction, $w_p$) of the two models fitted to data generated by the model with reset (left) and to data generated by the model without reset (right).

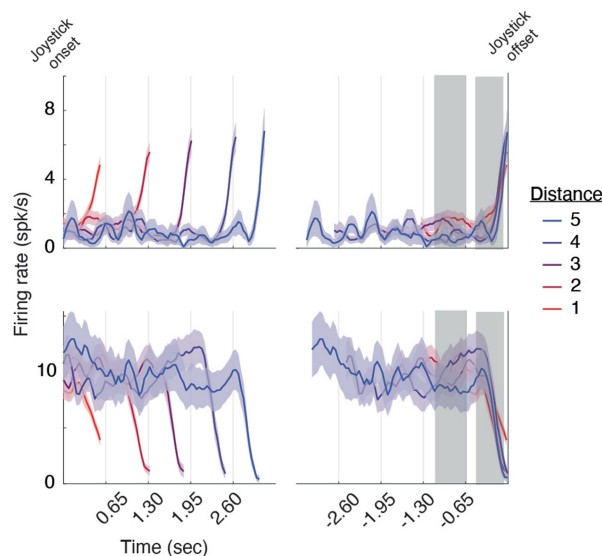

**Extended Data Fig. 10 | Motor preparation signals before joystick offset.**
Two example non-periodic neurons in the EC showing motor-related activity
before joystick offset similar to 'memory-trace' cells in Qasim et al.[23]. Overall,
the majority of neurons in both monkeys showed such motor-related response,
77% (477/614) in monkey A and 70% (607/864) in monkey M (two-sample *t*-test
(p < .0001) between average activity in the two intervals indicated by the
shaded grey regions). Within the non-periodic population, the proportion of
neurons with motor-related response was 81% (368/452) in monkey A and 89%
(432/481) in monkey M.

# Reporting Summary

## Statistics

For all statistical analyses, confirm that the following items are present in the figure legend, table legend, main text, or Methods section.

| n/a | Confirmed | |
|---|---|---|
| ☐ | ☒ | The exact sample size (*n*) for each experimental group/condition, given as a discrete number and unit of measurement |
| ☐ | ☒ | A statement on whether measurements were taken from distinct samples or whether the same sample was measured repeatedly |
| ☐ | ☒ | The statistical test(s) used AND whether they are one- or two-sided *Only common tests should be described solely by name; describe more complex techniques in the Methods section.* |
| ☐ | ☒ | A description of all covariates tested |
| ☐ | ☒ | A description of any assumptions or corrections, such as tests of normality and adjustment for multiple comparisons |
| ☐ | ☒ | A full description of the statistical parameters including central tendency (e.g. means) or other basic estimates (e.g. regression coefficient) AND variation (e.g. standard deviation) or associated estimates of uncertainty (e.g. confidence intervals) |
| ☐ | ☒ | For null hypothesis testing, the test statistic (e.g. *F*, *t*, *r*) with confidence intervals, effect sizes, degrees of freedom and *P* value noted *Give P values as exact values whenever suitable.* |
| ☐ | ☒ | For Bayesian analysis, information on the choice of priors and Markov chain Monte Carlo settings |
| ☒ | ☐ | For hierarchical and complex designs, identification of the appropriate level for tests and full reporting of outcomes |
| ☒ | ☐ | Estimates of effect sizes (e.g. Cohen's *d*, Pearson's *r*), indicating how they were calculated |

*Our web collection on statistics for biologists contains articles on many of the points above.*

## Software and code

Policy information about availability of computer code

| Data collection | Behavioral console was controlled via MWorks (mworks.github.io, version 0.9). Eye movements were collected using Eyelink 1000 (SR Research Ltd, Ontario, Canada). Joystick/Hand movements were collected using (PCIe6251 board, National Instruments, TX). 32- and 64-channel laminar probes (V-probe, Plexon Inc., TX) were used for extracellular recordings. A motorized micro-manipulator was used for targeting the region of interest (Narasighe Inc.). Recorded signals were preprocessed using OpenEphys (OpenEphys Inc., Lisbon, Portugal). Kilosort 2.0 was used for spike sorting. A python-based GUI (phy) was used to verify spike sorting quality manually. |
|---|---|
| Data analysis | Data analysis was performed using Matlab. |

For manuscripts utilizing custom algorithms or software that are central to the research but not yet described in published literature, software must be made available to editors and reviewers. We strongly encourage code deposition in a community repository (e.g. GitHub). See the Nature Portfolio guidelines for submitting code & software for further information.

## Data

Policy information about availability of data

All manuscripts must include a data availability statement. This statement should provide the following information, where applicable:
- Accession codes, unique identifiers, or web links for publicly available datasets
- A description of any restrictions on data availability
- For clinical datasets or third party data, please ensure that the statement adheres to our policy

| The data used to generate the associated figures and the URL for access will be made available on a public repository. |
|---|

# Field-specific reporting

Please select the one below that is the best fit for your research. If you are not sure, read the appropriate sections before making your selection.

☒ Life sciences          ☐ Behavioural & social sciences          ☐ Ecological, evolutionary & environmental sciences

For a reference copy of the document with all sections, see nature.com/documents/nr-reporting-summary-flat.pdf

# Life sciences study design

All studies must disclose on these points even when the disclosure is negative.

| | |
|---|---|
| Sample size | We used n=2 monkeys |
| Data exclusions | We did not exclude any data relevant to this study. We have included all cases when specific aspects of the data was used in a figure or analysis. |
| Replication | All results reported were replicated by cross-validation across different portions of the data. In all cases where the data was pooled across conditions, the analysis was also performed on individual conditions to ensure the results were robust. |
| Randomization | All experimental conditions (trials and trial types) were presented randomly. |
| Blinding | The trial conditions were randomly interleaved and therefore both the experimenters and the subjects were blind to the experimental conditions/groupings. |

# Reporting for specific materials, systems and methods

We require information from authors about some types of materials, experimental systems and methods used in many studies. Here, indicate whether each material, system or method listed is relevant to your study. If you are not sure if a list item applies to your research, read the appropriate section before selecting a response.

## Materials & experimental systems

| n/a | Involved in the study |
|---|---|
| ☒ | ☐ Antibodies |
| ☒ | ☐ Eukaryotic cell lines |
| ☒ | ☐ Palaeontology and archaeology |
| ☐ | ☒ Animals and other organisms |
| ☒ | ☐ Human research participants |
| ☒ | ☐ Clinical data |
| ☒ | ☐ Dual use research of concern |

## Methods

| n/a | Involved in the study |
|---|---|
| ☒ | ☐ ChIP-seq |
| ☒ | ☐ Flow cytometry |
| ☒ | ☐ MRI-based neuroimaging |

## Animals and other organisms

Policy information about studies involving animals; ARRIVE guidelines recommended for reporting animal research

| | |
|---|---|
| Laboratory animals | 2 rhesus macaque monkeys (Macaca mulatta). |
| Wild animals | Study did not involve wild animals. |
| Field-collected samples | Study did not involve field-collected samples. |
| Ethics oversight | All procedures were performed in compliance with the guideline of National Institutes of Health and the American Physiological Society, and approved by the MIT Committee on Animal Care. |

Note that full information on the approval of the study protocol must also be provided in the manuscript.

