## [Peer Review file · Nature]

Manuscript Title: Vector production via mental navigation in the entorhinal cortex

Reviewer Comments & Author Rebuttals

Reviewer Reports on the Initial Version:

Referees' comments:

Referee #1 (Remarks to the Author):

Summary

The authors examine single cell recordings from the entorhinal cortex of two non-human primates performing a mental navigation task in a 1d environment. In this task, the animals learned to deflect a joystick in the correct direction for a specific duration needed to move between two previously observed visual landmarks at a constant speed. Importantly, the animals were trained on a subset of initial and target landmark pairs but showed rapid generalisation to new landmark pairs during later test trials, consistent with the use of a structured task representation or 'cognitive map'.

Focussing on periods of joystick deflection during mental navigation, the authors find that a significant proportion of recorded neurons show periodic activity patterns with a periodicity that matches the temporal spacing between landmarks (which are not visible during this part of the task). This periodicity appears to correlate with task performance, being reduced during error trials. In addition, they show that a second population of neurons show downward ramping responses during mental navigation, most often with a constant rate of decay during the production of different length vectors (referred to here as an 'offset code').

These experiments aim to address a topic of considerable current interest (the role of entorhinal grid cells in mental navigation), the data are fairly unique, and the combination of modelling and empirical work is to be commended. The manuscript is clearly written and I have no concerns about the methodology, use of statistics, or credit of previous work. The presence of periodic firing patterns could also be of significant interest, particularly if those firing patterns can be linked directly with behaviour. There is little to convince me that these arise from entorhinal grid cells, however, while analyses of the relationship with task performance are coarse and could be equally well explained by various alternative hypotheses. It seems more likely that the periodic responses are from 'landmark neurons', as described in the modelling work here and in previous rodent entorhinal cortex recordings (e.g. their ref 34). As such, despite the rigorous analysis and accompanying theory, I am not sure the authors conclusions are well supported, or that these results would be of immediate interest to researchers in my field or beyond.

Major Suggested Improvements

[1] As the authors state, "a key feature of grid cell dynamics ... is a broad distribution of relative firing phases across cell pairs that is conserved across conditions". However, their results do not convincingly show either feature – the cross-correlations in Fig 3E and Fig S6 seem to show that firing phases are clustered, and data from adjacent task periods does not clearly constitute a

different condition. In fact, there are several other aspects which suggest to me that these periodic firing patterns are more likely to be 'landmark neurons' as seen in rodent entorhinal cortex (their ref 34); in human entorhinal cortex (Qasim et al., Nature Neuroscience 2019); and in the modelling work presented here. Primarily, the fact that the periodicity exactly matches the (constant) distance between adjacent landmarks in this task. Several steps could be taken to make a more convincing case that these neurons are grid cells. Ideally, this would incorporate a version of the task with an irregular distribution of landmarks (as acknowledged in the Discussion), but I appreciate that this is not possible to address with the current data set.

First, the authors could address whether the distribution of relative phases across cell pairs in their data is (non-)uniform; and whether it more closely resembles the distribution of relative phases across pairs of grid cells or landmark neurons in their model. Second, if possible, they could address whether this distribution is conserved outside of this task (rather than in ITI periods within the task, particularly given that the cells show reduced periodicity during those periods). Third, they could better characterise the firing patterns of these neurons in relation to known properties of entorhinal grid cells. For example: do these neurons show periodic firing patterns from the first learning trial, and across movement in both directions (the differing cell counts in Fig S3 suggest not)? Finally, if the periodic neurons are grid cells that receive an error correcting input from landmark cells, then variability in their firing should also be periodically reduced during mental navigation (like the 'model with reset' illustrated in Fig 4E) – is this the case?

[2] A more nuanced analysis that connected the periodicity of task-modulated neurons to behaviour would be of great benefit. If I understand the task design correctly, there is a fairly wide temporal window for 'correct' responses (i.e. during which the target landmark is closest to the current position when the joystick is released). Can the authors sub-select correct trials (or groups of correct trials) where responses fall early vs late within this window, and show that periodicity is shorter or longer, respectively? I suspect that the results shown in Fig 3E, F are consistent with this outcome, but less intuitive. Similarly, is it of any significance that the autocorrelation peak shown in Figure 2D for Monkey A is slightly below 0.65s, matching the fact that this animal seems to consistently respond slightly ahead of time (as shown by a regression slope < 1 in Fig 1B); whereas the peak for Monkey M is closer to 0.65s, and this animal seems to consistently respond more accurately (as shown by a regression slope ~ 1 in Fig 1B)?

Minor Comments

[1] The figures could be clearer. The example in Fig 1A shows the joystick being deflected to the right, and the arrow under 'Joystick onset' points to the right, but the terminal state appears to be to the left of the initial state in the linear map shown below, which may cause some confusion. Similarly, in the example shown in Fig S1A, the joystick is being deflected to the right, and the arrow under 'Joystick onset' points to the right, but location on the map appears to be moving to the left. In Fig 1E, it is not clear to me what the 'regression slope ≥ 0.8 ' text refers to. It does not appear that the final regression slope values of the training sessions are ≥ 0.8 for either animal - please clarify. There is a typo in Fig 2E ('3rd peak / shuffled'). In Fig 3B, it is confusing to show results with negative R^2 values (which are impossible) – it would be preferable to change the labels to 'cross-validated R^2 ' or similar. In Fig 3C, what would the third dimension (D3) correspond to? What is the difference

between Fig 3E and 3F? Does the former show data for a single neuron, and the latter for all recorded neurons? Please specify

[2] Is it worth using a binomial test to demonstrate that the proportion of periodic neurons identified in each animal (i.e. 99/614 and 74/864) is greater than expected by chance?

[3] How many downward ramping neurons were recorded? Similarly, how many neurons showed upward ramping responses? Are there more than expected by chance? How did each of these populations overlap with periodic neurons? I cannot see these numbers anywhere in the manuscript

[4] Following the previous point, several general details about the data set are required. How many task trials were completed in each session? Were the landmark images constant across all sessions? How much task experience did each animal have during the neural recording sessions? Fig 1C suggests that the animals completed 101 and 181 sessions, respectively, but Fig 1E shows ~60 sessions per animal, and the Methods state that there were 32 neural recording sessions – how do these values correspond to one another? Were neurons recorded in each behavioural session treated as independent, or was the recording apparatus left in place and the same neurons recorded across multiple behavioural sessions?

[5] How do the authors cross-validate the downward ramping r squared values? Does this include data from all trials? I cannot see these details in the Methods (but apologies if I have missed them)

[6] How might offset coding support mental navigation? I find it difficult to envisage how a population of neurons that exhibit offset coding would allow location on the linear track to be decoded. Perhaps the authors could briefly explore this

[7] There is a typo on line 510, in the 'Analysis of neural data' section of the Methods: 'trail-averaged' should be 'trial averaged'

Referee #3 (Remarks to the Author):

Neupane et al. describe the temporally periodic firing of non-human primate EC neurons locked to mental representations of landmarks along a linear path. This could be an excellent addition to the literature and significantly improve our understanding of the function of the EC.

The originality and significance are good. Data and methodology appear strong. Statistics are appropriate. The conclusions are robust and validated with a very nice GC+LM model. All aspects of the text are clear and concise. The citations of the literature were nicely done.

Minor concerns:

1. The figures need work—lines are too thin, shaded patches are too faint (increase the alpha value), the panels are too small, and the black-to-red graded color scheme makes it extremely difficult to discern the curves. I had to zoom in very far on almost every plot to assess the data. Please fix these issues.

2. I am troubled that the description of electrophysiology is so thin. First, it's unclear what electrodes were used—these should all be clear: brand name or made in-house, materials, site geometry, and impedance. Second, the description of the spike sorting is poor (Purely automated? What manual supervision was done? Do you have quality statistics for the clusters?). Finally, it's unclear whether multi-units were included in the subsequent analyses. I don't see how the inclusion of multi-units would be beneficial. Please provide waveforms for some of the single units and describe the quality of the population of units that went into the analyses.

Author Rebuttals to Initial Comments:

We thank the reviewers for their careful reading of our manuscript and insightful comments. Here, we provide an overview of how we have addressed the comments.

Referee #1 (Remarks to the Author):

Summary

The authors examine single cell recordings from the entorhinal cortex of two non-human primates performing a mental navigation task in a 1d environment. In this task, the animals learned to deflect a joystick in the correct direction for a specific duration needed to move between two previously observed visual landmarks at a constant speed. Importantly, the animals were trained on a subset of initial and target landmark pairs but showed rapid generalisation to new landmark pairs during later test trials, consistent with the use of a structured task representation or 'cognitive map'.

Focussing on periods of joystick deflection during mental navigation, the authors find that a significant proportion of recorded neurons show periodic activity patterns with a periodicity that matches the temporal spacing between landmarks (which are not visible during this part of the task). This periodicity appears to correlate with task performance, being reduced during error trials. In addition, they show that a second population of neurons show downward ramping responses during mental navigation, most often with a constant rate of decay during the production of different length vectors (referred to here as an 'offset code').

These experiments aim to address a topic of considerable current interest (the role of entorhinal grid cells in mental navigation), the data are fairly unique, and the combination of modelling and empirical work is to be commended. The manuscript is clearly written and I have no concerns about the methodology, use of statistics, or credit of previous work. The presence of periodic firing patterns could also be of significant interest, particularly if those firing patterns can be linked directly with behaviour. There is little to convince me that these arise from entorhinal grid cells, however, while analyses of the relationship with task performance are coarse and could be equally well explained by various alternative hypotheses. It seems more likely that the periodic responses are from 'landmark neurons', as described in the modelling work here and in previous rodent entorhinal cortex recordings (e.g. their ref 34). As such, despite the rigorous analysis and accompanying theory, I am not sure the authors conclusions are well supported, or that these results would be of immediate interest to researchers in my field or beyond.

We thank the reviewer for this comment. We agree that landmark cells and grid cells are not clearly dissociable in an experiment with regular landmarks. Indeed, when an animal traverses a 1D track in the presence of visible regular landmarks, we would expect landmark cells – that receive sensory input from visible landmarks – to have regular firing fields and thus look like grid cells. However, one critical aspect of our experiment that heavily influenced our thinking was that neurons had regular firing fields even though the landmarks were invisible. In this setting, the regular firing fields cannot be due to external input and must instead be generated through recurrent attractor dynamics, which is a prominent feature of grid cells in EC. The question remains, however, as to whether the periodic neurons we recorded inherit their firing properties from attractor dynamics of grid cells or if they are part of a separate landmark cell population that generates similar attractor dynamics on its own.

Guided by the reviewer's comments, we have performed several analyses to address this question. Our analyses suggest that the periodicity likely originates from cells whose connectivity establishes strong attractor dynamics suggesting that they are functionally homologous to grid cells.

Major Suggested Improvements

[1] As the authors state, "a key feature of grid cell dynamics ... is a broad distribution of relative firing phases across cell pairs that is conserved across conditions". However, their results do not convincingly show either feature – the cross-correlations in Fig 3E and Fig S6 seem to show that firing phases are clustered, and data from adjacent task periods does not clearly constitute a different condition. In fact, there are several other aspects which suggest to me that these periodic firing patterns are more likely to be 'landmark neurons' as seen in rodent entorhinal cortex (their ref 34); in human entorhinal cortex (Qasim et al., Nature Neuroscience 2019); and in the modelling work presented here. Primarily, the fact that the periodicity exactly matches the (constant) distance between adjacent landmarks in this task. Several steps could be taken to make a more convincing case that these neurons are grid cells. Ideally, this would incorporate a version of the task with an irregular distribution of landmarks (as acknowledged in the Discussion), but I appreciate that this is not possible to address with the current data set.

As highlighted by the reviewer, there have been reports of landmark neurons in EC, neurons that are activated by the presence of visible landmarks in the arena. In our experiment, the landmarks were not visible. However, it is possible that landmark cells form a distinct recurrent network capable of generating attractor dynamics to produce endogenous activity in the absence of visible landmarks with no dependence on grid cells. To address this question, we considered the two papers that the reviewer highlighted.

Kinkabwala et al. 2020 found landmark cells in EC of mice associated with tower-like visual cues along a linear track in virtual reality. We requested data from their landmark neurons (Figure 2 of their paper) and applied our cell-cell correlation analysis to test if they exhibit hallmarks of attractor dynamics observed in our data as well as grid cells (preserved correlation structure). We found that cross-correlations were preserved across epochs with visible landmarks but abolished when the landmarks were invisible (Fig R1; H_0 : correlation for region B visible = correlation for region B invisible; one-tailed 2 sample t-test(198)=13.73, $p \ll \ll .0001$). This result indicates that landmark neurons do not exhibit the hallmarks of recurrent dynamics that enable endogenous periodic activity, as previously reported for rodent grid cell populations (Trettel et al. 2019; Gardner et al. 2019). We also note that, unlike our results, landmark neurons in Kinkabwala et al. 2020 were not modulated when landmarks were absent. Together, these findings reject the hypothesis that the periodicity in our neurons comes from landmark neurons like those reported in Kinkabwala et al. We now mention this in the main paper and have added this analysis and the figure as a supplement to our discussion.

Qasim et al. 2019 recorded EC neurons in humans in a task that required moving along a virtual linear track and pressing a button to stop at the location of an invisible (memorized) landmark. This study found a group of so-called memory-trace cells in EC (as well as the hippocampus and cingulate cortex) that were activated right before participants pressed the button. The relevance of this study compared to Kinkabwala et al. 2020 is that, in this study, the landmark, where subjects were instructed to press a button, was invisible.

cross-correlation analysis on data from
Kinkhabwala et al (eLife 2020)

Fig R1.

a. Reproduced figure 2.a. from Kinkhabwala et al. 2020 showing the firing pattern of a landmark cell when landmarks were visible (top) and invisible (bottom).

b. Cell-cell spike time cross-correlation across all the simultaneously recorded cell pairs for region B (left) and for region A (right), both sorted according to region B data. The bottom plot shows the same analysis performed on data for Region B when landmarks were invisible.

c. Cross-correlation values averaged over the lag of +5ms for data from region B when landmarks were visible and plotted against the values from region A (left) and region B with landmarks invisible (right).

Unfortunately, we were unable to obtain the human neurophysiology data from Qasim et al. due to data privacy issues. Therefore, we cannot verify whether cell-cell correlations in their data were preserved across epochs and conditions. Absent direct access to the data, we think that Qasim et al. results are consistent with two interpretations. First, the response modulations may be related to motor preparation. In their experiment, participants had to press a button when they reached an invisible landmark. As such, the memory trace for the landmark was confounded by the motor response. Notably, a similar motor preparation signal was evident in our dataset as well near the end of the trial at the time of joystick offset. However, this signal was present regardless of whether neurons had periodic activity or not (Fig R2), suggesting that the two populations are not the same. We have added these points to our discussion and thank the reviewer for pointing out this relevant paper.

Fig R2. Two example non-periodic neurons in EC showing motor-related activity prior to joystick offset similar to ‘memory-trace’ cells in Qasim et al 2019. Overall, the majority of neurons in both animals showed such motor-related response (77% (477/614) in monkey A and 70% (607/864) in monkey M; 2-sample t-test ($p < .0001$) between activity pooled across trials in 2 time windows indicated by the shaded gray regions). Within the non-periodic population, the proportion of neurons with motor-related response was 81% (368/452) in monkey A and 89% (432/481) in monkey M.

First, the authors could address whether the distribution of relative phases across cell pairs in their data is (non-)uniform; and whether it more closely resembles the distribution of relative phases across pairs of grid cells or landmark neurons in their model.

This is an excellent suggestion. As the reviewer has surmised, when measuring spontaneous activity, the relative phase distribution of grid cells is expected to be uniform. Given that the relative phase distribution across our periodic neurons is non-uniform, this raises the question of whether the periodic neurons are part of the grid cell network. Two factors underlie this apparent discrepancy. First, the relative phase distribution of grid cells under spontaneous activity (uniform) differs from the distribution when grid cells interact with landmark cells (non-uniform). Following the reviewer’s suggestion, we used simulations of our model to demonstrate this point.

Fig R3 shows the sorted cross-correlogram of all the neurons with respect to one neuron in the population. In one scenario (Figure R3a, bottom), we removed the interaction between the grid cells and landmark cells. In this case, the relative phases span the range perfectly linearly (Fig R3a, bottom left) and have a uniform distribution (Fig R3a, bottom right). In the other scenario, we simulated the grid cells in the presence of landmark cells (Figure R3a, top). In this case, input from landmark cells has a reset-like effect on the phase of the grid cells, which distorts the phase space (Fig R3a, top left) and makes the distribution non-uniform (Fig R3a, top right). In other words, the non-uniformity of phase distribution in our dataset is perfectly consistent with a grid cell network whose connections to landmark neurons are modified through Hebbian-like learning.

The second factor that makes the phase distribution appear non-uniform is uninteresting and technical. In general, phase distribution can be analyzed in two ways. One approach is to compare the phases of all grid cells

in the population to one reference grid cell. This is the approach we used in Figure R3a. This approach is unbiased and correctly reveals the uniformity of phase distribution under spontaneous activity. An alternative approach is to compute the phase distribution using all pairs of grid cells in the population. This approach is sometimes adopted to increase statistical power but it is a biased measure of phase distribution and leads to non-uniform distributions simply because in an ordered set of items, the shortest relative distance is most frequent, and the longest relative distance is least frequent. Similarly, the distribution of the relative phase is biased towards 0 because the cell pairs with nearby phases are more frequent. Comparing FigR3a and R3b makes this point clear, especially in the case of without landmarks (Fig R3a bottom vs. R3b bottom). Together, these results demonstrate that the observed phase distribution in our data is consistent with a model of grid cells that interact with landmark cells.

Fig R3.

a. Left. Cross-correlation structure of pairs of units in the model with respect to one reference unit with reset (top) and without reset (bottom). Right. Distribution of the lag where peak phase occurs in the CAN model with reset (top) and without reset (bottom).

b. Same as (a) but across all unit pairs.

Second, if possible, they could address whether this distribution is conserved outside of this task (rather than in ITI periods within the task, particularly given that the cells show reduced periodicity during those periods).

Unfortunately, we did all our recordings in EC from animals during the mental navigation task and thus do not have data from outside the task or prior to training. However, we wish to be as responsive as possible within the bounds of what we can do. As such, we extended our analysis beyond the ITI period and looked at all task epochs, including the inference epoch (before mental navigation), error trials, and for both directions of movement (in addition to ITI). We found that the cell-cell correlations were preserved across all epochs. (Fig R4a). Here, we wish to draw attention to two epochs, ITI and the inference epoch (1400ms window after the onset of the start landmark image).

The ITI epoch is relevant because, in this interval, animals are not engaged in any task-relevant computations. Therefore, the preserved cell-cell correlations in this epoch suggest that the neurons were part of the grid cell network.

The inference epoch is relevant for testing an alternative hypothesis: that our recordings were from landmark cells and that the preserved cell-cell correlations were inherited from inputs landmark cells receive from grid cells. In the inference epoch, the animals view two visible images associated with the start and the end landmarks. If the neurons we recorded were landmark cells, they should receive strong visual input from these visible landmarks. In other words, during the inference epoch, in addition to the intrinsic input from grid cells, landmark cells would receive strong extrinsic input from visible landmarks. Because of this additional input, the cell-cell correlation between landmark cells should drop during the inference epoch. To understand the logic behind this prediction, consider the extreme case in which the activity of landmark cells is completely dominated by the extrinsic input. In such a scenario, we expect the cell-cell correlation during the inference epoch to be completely abolished by the strong extrinsic input. More generally, in the presence of external input, we expect cell-cell correlations to drop. However, the data was inconsistent with this prediction. We found that the cross-correlation patterns were preserved in this epoch (Fig R4b,c; $r = 0.93$, $p < .0001$ for monkey A and $r = .83$, $p < .0001$, for monkey M), and there was no significant decrease in the magnitude of cell-cell correlations during the inference epoch compared to the navigation epoch (right-tailed 2 sample t -test $2(418) = 1.4928$, $p = 0.068$ for monkey A and t -test $2(592) = -9.1$, $p = 1$ for monkey M, Figure R4). This result suggests that our recordings were from cells with strong recurrent connectivity, similar to grid cells. We have included these additional analyses and the corresponding figure (Figure S8) in the updated results section.

Fig R4.

a. Cross-correlation structure of cell pairs in the window during mental navigation (first panel) and during the presentation of start and target landmarks (second panel) sorted according to the peak correlation lag in the mental navigation window. Similarly, cross-correlation structure in mental navigation epoch for error trials (third panel) and right direction panel (fourth panel). Note that not all neurons were stable towards the end of the session when we ran visible trials, and therefore, there are several rows with no data.

b. Cross-correlation values averaged over the lag of ± 5 ms for data from mental navigation plotted against data from ITI (left, top) and from the image presentation window (right, top). Bottom, same as the top, but for a session where the neurons were non-periodic.

c. Same as (b) for monkey M.

Third, they could better characterise the firing patterns of these neurons in relation to known properties of entorhinal grid cells. For example: do these neurons show periodic firing patterns from the first learning trial, and across movement in both directions (the differing cell counts in Fig S3 suggest not)?

Unfortunately, we did not record neural data during the learning sessions. We, therefore, cannot make definite statements about the patterns of activity before and during learning. Please note that we started this project with a great deal of uncertainty about whether we would find task-relevant modulations in the primate EC, let alone the discovery of such compelling periodicity (there is no precedence for temporal periodicity in primate EC). However, having seen the results, we are now excited to redo the experiment and record from EC throughout learning. We are motivated to look at EC during learning because, according to our model, this periodicity emerges as a result of Hebbian-like learning. In other words, we expect the periodicity to co-evolve with the animal's understanding of the temporal structure of the task. We are also excited to investigate how the brain might adapt this periodic activity in novel conditions, including mental navigation under different speeds and mental navigation in the presence of non-periodic landmark organization. Our model makes specific predictions for each of these conditions that we hope to test in future experiments.

Regarding the second question about directionality, we have revised the manuscript and included a full characterization of periodicity and ramping as a function of direction tuning across our EC population. Briefly, we found a diversity of neurons, with some showing similar characteristics across directions and others with direction-dependent properties. As shown in Table TS1, of all periodic neurons, 24% were direction-invariant in monkey A and 22% in monkey M. These numbers are consistent with rodent grid cells that are not conjunctive with head direction tuning. However, we don't have access to laminar information to compare our numbers against more fine-grained anatomical information that exists in the rodent grid cell literature (Sargolini et al. 2006). We have added the table as supplementary to the Results section.

Table TS1: Periodic and ramping cell proportions across two directions

	Property	Left only	Right only	Both directions	Total
Monkey A	Periodicity	90 (39%)	85(37%)	56 (24%)	231/614
	Ramping	27 (46%)	23 (40%)	8 (14%)	58/614
Monkey M	Periodicity	142 (46%)	101(32%)	68 (22%)	311/864
	Ramping	40 (39%)	28 (27%)	33 (33%)	101/864

Finally, if the periodic neurons are grid cells that receive an error-correcting input from landmark cells, then variability in their firing should also be periodically reduced during mental navigation (like the 'model with reset' illustrated in Fig 4E) – is this the case?

We really appreciate this question because it has led to a new discovery in our data, which we wouldn't have made without the reviewer's comment. If/when this paper gets approved for publication, we would be delighted to acknowledge this important contribution from the anonymous reviewer if they are not opposed to it.

We reasoned that the simplest measure of variability would be the Fano factor: variance of spiking activity divided by the mean. Usually, this variance grows with the mean, similar to a Poisson process that has a Fano factor of unity. We reasoned that if the periodicity in EC neurons has an error-correcting function (i.e., reducing variability), the Fano factor should be modulated with the same periodicity independent of the neurons' firing rate (i.e., independent from the Poisson-like variability).

Surprisingly (and in line with the reviewer's intuition), we found many neurons that exhibited a periodic Fano factor at the periodicity of 0.65 sec. This was evident in both periodic and non-periodic neurons, i.e., both when the mean firing rate was and was not periodic (Fig R5a,d). To quantify this observation, we first created a null distribution for each neuron by passing its mean firing rate through a Poisson process 100 times and used this surrogate data to generate a distribution of Fano factors under the *Poisson Null*. We also estimated the distribution of the observed Fano factors by bootstrapping the spiking data 100 times. We then computed the autocorrelation of the Fano factor and the corresponding periodicity index for both the actual and the null distributions. Next, we estimated the periodicity of each neuron whose Fano factor had a significantly higher periodicity index in the window of 300 to 900 ms lag, compared to its corresponding *Poisson Null* (2-sample t-test with $p < (0.0001/20)$, with Bonferroni correction). Fig R5c,f show that periodicities of the neurons with significant periodic Fano factor were tightly clustered at the behaviorally relevant periodicity of 0.65 sec. The proportion of neurons with periodic Fano factor was 43% (378/864) in Monkey M and 55% (341/614) in Monkey A. This result provides evidence that the observed periodicity in EC is associated with an error-correcting function during mental navigation, in line with the reviewer's expectation. We are excited by this finding and plan to devise models and design experiments to understand the computational basis of error correction in EC. We have included this finding in a new supplementary Fig S11.

Fig R5.

a. Two example neurons for monkey A showing periodic pattern in fano factor (bottom) but little or no periodicity in average spike count time series (top).

b. Autocorrelogram of Fano factor for the two example neurons (blue) and their corresponding (Poisson) null autocorrelogram (orange).

c. Distribution of periodicity of the Fano factor for neurons with significant periodicity compared to their corresponding (Poisson) null data. Dotted lines denote the window within which significant difference was tested.

d,e,f. Same as (a,b,c) for monkey M.

[2] A more nuanced analysis that connected the periodicity of task-modulated neurons to behavior would be of great benefit. If I understand the task design correctly, there is a fairly wide temporal window for 'correct' responses (i.e., during which the target landmark is closest to the current position when the joystick is released). Can the authors sub-select correct trials (or groups of correct trials) where responses fall early vs late within this window and show that periodicity is shorter or longer, respectively? I suspect that the results shown in Fig 3E, F are consistent with this outcome, but less intuitive.

We thank the reviewer for this suggestion. We performed the analysis suggested by the reviewer and observed a small but significant increase in the periodicity (i.e., lower frequency) on trials when the animals overshot the target landmark (Fig R6). We have now added this result in the updated results section as Supplementary Figure S6.

Fig R6. We focused our analysis on trials with a distance of 3, which were long enough to quantify periodicity (i.e., two full periods) and afforded enough repeats to gain statistical power. For these trials, we sorted the distribution of timing error in ascending order and pooled trials from the top and the bottom quartiles representing undershoot and overshoot trials. We then computed the periodicity for the two groups of trials following the same auto-correlogram procedure as in Fig 2. Note that the number of neurons is smaller than the total number of periodic neurons because neurons with less than 15 trials in each bin were dropped from the analysis.

a. Left. An example neuron's raster whose top (black) and bottom quartile (red) of timing error trials, representing undershoot (black) and overshoot (red), respectively, were compared. Middle. Autocorrelogram of undershoot and overshoot trials. Right. Monkey A: Periodicity index at various lags for undershoot and overshoot trials show high periodicity for overshoot trials.

b. Left. Scatter plot of all periodic neuron's periodicity for undershoot trials against those for overshoot trials. Inset: Distribution of periodicity difference. Same as (left) for monkey M. Across neurons, the response period was significantly larger for overshoot trials in both animals (Fig R5b; rank-sum test; $Z(194)=-5.80$, $p<<<.0001$ for monkey A and $Z(161)=-3.3$, $p<<<.0001$ for monkey M).

Similarly, is it of any significance that the autocorrelation peak shown in Figure 2D for Monkey A is slightly below 0.65s, matching the fact that this animal seems to consistently respond slightly ahead of time (as shown by a regression slope < 1 in Fig 1B); whereas the peak for Monkey M is closer to 0.65s, and this animal seems to consistently respond more accurately (as shown by a regression slope ~1 in Fig 1B)?

Thank you for pointing out this observation. As noted, the plot in Fig 1B shows behavior for one session in each animal in which the regression slope for one animal is closer to 1. However, please note that the deviation of the regression slope from unity is markedly influenced by the longest distance for which we have the least number of trials per session. Indeed, as shown in Fig. 1C, the regression slope varied across sessions and animals. Therefore, the more appropriate analysis is one that compares periodicity across trials shown above, as was suggested by the reviewer.

Minor Comments

[1] The figures could be clearer. The example in Fig 1A shows the joystick being deflected to the right, and the arrow under 'Joystick onset' points to the right, but the terminal state appears to be to the left of the initial state in the linear map shown below, which may cause some confusion. Similarly, in the example shown in Fig S1A, the joystick is being deflected to the right, and the arrow under 'Joystick onset' points to the right, but location on the map appears to be moving to the left. In Fig 1E, it is not clear to me what the 'regression slope ≥ 0.8 ' text refers to. It does not appear that the final regression slope values of the training sessions are ≥ 0.8 for either animal - please clarify. There is a typo in Fig 2E ('3rd peak / shuffled'). In Fig 3B, it is confusing to show results with negative R^2 values (which are impossible) – it would be preferable to change the labels to 'cross-validated R^2 ' or similar. In Fig 3C, what would the third dimension (D3) correspond to? What is the difference between Fig 3E and 3F? Does the former show data for a single neuron, and the latter for all recorded neurons? Please specify

We apologize for the confusion. In our task design, a rightward joystick deflection drifts the sequence rightward, bringing a terminal state originally in the left to the center. One way to think about this is that the joystick moves the environment – not the subject. This may have caused confusion because, in real navigation, one moves toward the destination. We designed the task that way so that the sequence length is not limited by the size of the screen.

We used the regression slope of 0.8 as a training criterion. After the animals reached this criterion in at least one session, we continued the training for one more week (7 sessions) to ensure stable behavior (at least one other session where the animal reached the criterion) before testing the animals on generalization pairs. The slope is below 0.8 in the last session due to variability in animal behavior across sessions. We have added a section called *performance criterion* in the Methods to clarify this point.

The typo in Fig2E is now fixed. Thank you.

We have now labeled the R^2 values as 'cross-validated R^2 '.

The third dimension, D3, is simply a hypothetical arbitrary dimension presented to depict a high-dimensional neural space.

Does the reviewer mean 2E and 2F? Yes, 2E is for a single neuron example, and 2F is for all periodic neurons (not all recorded neurons). We have now added labels to the panels.

[2] Is it worth using a binomial test to demonstrate that the proportion of periodic neurons identified in each animal (i.e. 99/614 and 74/864) is greater than expected by chance?

They are greater than expected by chance (binomial test at 5%, $pval < 1e-5$). We now report that the number of periodic neurons is 231/614 in monkey A and 311/864 in monkey M (see Table TS1 for more details).

[3] How many downward ramping neurons were recorded? Similarly, how many neurons showed upward ramping responses? Are there more than expected by chance? How did each of these populations overlap with periodic neurons? I cannot see these numbers anywhere in the manuscript

Thank you for this comment. We have now tabulated the cell count systematically in terms of ramping (down vs. up), periodicity, and directionality and included it as Supplementary information in the Results section.

Table TS2: Proportions of cells showing upward ramp, downward ramp, ramping only, periodicity only, or both ramping and periodicity across two directions.

	downward	upward	Ramping only	Periodic only	Both
Right, Monkey A	16	15	29	139	2
Left, Monkey A	32	3	31	142	4
Right, Monkey M	4	57	46	154	15
Left, Monkey M	14	59	58	195	15

[4] Following the previous point, several general details about the data set are required. How many task trials were completed in each session? Were the landmark images constant across all sessions? How much task experience did each animal have during the neural recording sessions? Fig 1C suggests that the animals completed 101 and 181 sessions, respectively, but Fig 1E shows ~60 sessions per animal, and the Methods state that there were 32 neural recording sessions – how do these values correspond to one another? Were neurons recorded in each behavioural session treated as independent, or was the recording apparatus left in place and the same neurons recorded across multiple behavioural sessions?

We apologize for this lack of clarity. We have carefully revised the Methods section to clarify these questions. Briefly, we used Fig 1C to show behavior in all sessions and Fig 1E to show early sessions only to demonstrate generalization behavior. Animals were trained without a recording chamber. All recordings were carried out after the completion of training and verification that animals could generalize. We felt it was important to verify animals' ability to generalize as an important criterion for determining the suitability of the task before embarking on physiology. Once implanted, recordings were done acutely by lowering the electrode into EC at the beginning of each session. The recording electrode was retracted at the end of the session, and the procedure was repeated in subsequent sessions. Neurons recorded in each behavioral session are, therefore, independent. We have clarified these points in the revised Methods.

[5] How do the authors cross-validate the downward ramping r squared values? Does this include data from all trials? I cannot see these details in the Methods (but apologies if I have missed them)

Our cross-validated R^2 metric was calculated in the following way:

1. Divide all trials randomly into training (y_{train}) and test trials (y_{test}). For each group, estimate the mean firing rate from 200 ms before joystick offset or onset.
2. Estimate a linear regression slope for training data (dummy variable: $X=[1\ 2\ 3\ 4\ 5]$ for the five distances)
3. Calculate the predicted training data from the regression slope $y_{pred} = \text{Beta} * X$

4. Calculate cross-validated R^2 as variance accounted for by test data y_{test} :

$$R^2 = 1 - \frac{\Sigma (y_{\text{test}} - y_{\text{pred}})^2}{\Sigma (y_{\text{test}} - \underline{y_{\text{test}}})^2}$$

We have added subtitles to the Methods to facilitate tracking down this and related information in the Methods.

[6] How might offset coding support mental navigation? I find it difficult to envisage how a population of neurons that exhibit offset coding would allow location on the linear track to be decoded. Perhaps the authors could briefly explore this

The reviewer raises an important point about the nature of the neural code at joystick offset. Our results indicate that responses at that time encode relative distance and not the end image, which we now show more comprehensively in supplementary Fig S6 (also below, as Fig R7 a,b). This coding scheme is advantageous because it is not tied to a specific image set and can thus be readily used to navigate through linear tracks punctuated by other images, so long as the images are structured the same way. However, the more generalized distance coding we have found in EC comes at a cost. If the coding was in terms of images, the brain would be able to use neural activity at joystick offset to determine when to stop (i.e., when the endogenously encoded image matches the target image). With distance, such direct comparison is not possible. This problem can be solved in one of two ways. One possibility is to have a representation of the desired distance elsewhere in the brain and use that to compare to the EC firing rate to signal a match. The other possibility is to have concurrent representations of distance and image but in different brain areas, distance in EC, and image in a brain area that communicates with EC. We have followed the reviewer's suggestion and revised the Discussion to explore these two possibilities for decoding offset-coding.

However, independent of revisions to the manuscript, we would like to share with the reviewer an observation from the preliminary analysis of recordings of neural activity in the hippocampus (HC) of two animals, which is consistent with the latter solution (i.e., a representation of images at joystick offset). We have included these observations in the figure below (Fig R7 c,d; not included in the paper). The key finding is that, unlike EC, where the majority of neurons encoded relative distance, in HC, we found two seemingly non-overlapping subpopulations, one coding for relative distance and another for target image (note the bimodal distribution of relative variance explained by distance versus target image). While this is too preliminary to make a definite statement, based on this result, we think it is possible that EC and HC have complementary representations, an abstract and generalizable representation of relative distance in EC, and a more contextualized encoding of the images in HC. We hope to follow up on these results in the future to understand exactly how this complementary coding scheme in the medial temporal lobe is put to use for the control of behavior.

Fig R7.

a. Variance explained for start landmark factor plotted against variance explained for distance factor (blue: significant effect for distance, red: significant effect for start landmark, black: both, gray: none). Inset: distribution of the difference of variance explained, showing that the distance effect is significantly larger than the start landmark effect (2-sample t-test, $p < .0001$).

b. Same as a for target landmark factor vs. distance factor (blue: significant effect for distance, red: significant effect for target landmark, black: both, gray: none). Inset: distribution of the difference of variance explained, showing that the distance effect is significantly larger than the target landmark effect (2-sample t-test, $p < .0001$).

c,d. Same as a and b for neurons in the hippocampus. Distributions for HC neurons are bimodal compared to those for EC.

[7] There is a typo on line 510, in the 'Analysis of neural data' section of the Methods: 'trail-averaged' should be 'trial averaged'

Noted.

Referee #3 (Remarks to the Author):

Neupane et al. describe the temporally periodic firing of non-human primate EC neurons locked to mental representations of landmarks along a linear path. This could be an excellent addition to the literature and significantly improve our understanding of the function of the EC.

The originality and significance are good. Data and methodology appear strong. Statistics are appropriate. The conclusions are robust and validated with a very nice GC+LM model. All aspects of the text are clear and concise. The citations of the literature were nicely done.

Minor concerns:

1. The figures need work—lines are too thin, shaded patches are too faint (increase the alpha value), the panels are too small, and the black-to-red graded color scheme makes it extremely difficult to discern the curves. I had to zoom in very far on almost every plot to assess the data. Please fix these issues.

We apologize for the figures not being clear. Here is one example of a revised figure (Fig 2 from the paper). We have changed the color scheme to denote distances and increased the line size, the alpha values, and the size of some panels.

Revised Figure 2.

2. I am troubled that the description of electrophysiology is so thin. First, it's unclear what electrodes were used—these should all be clear: brand name or made in-house, materials, site geometry, and impedance. Second, the description of the spike sorting is poor (Purely automated? What manual supervision was done? Do you have quality statistics for the clusters?). Finally, it's unclear whether multi-units were included in the subsequent analyses. I don't see how the inclusion of multi-units would be beneficial. Please provide waveforms for some of the single units and describe the quality of the population of units that went into the analyses.

We apologize that this information was difficult to discern from the Methods. We have included a Table (see below) in the Methods to make sure the relevant information about physiology is readily accessible. Impedance measurements were made regularly with the OpenEphys recording system. All channels had an impedance of 275(+/-50) kOhms.

Electrodes from Plexon Inc.	Length (mm)	Diameter (μm)	Electrode spacing (μm)	Recording span (mm)
32-channel V-probe	100	240	100	3
64-channel V-probe	120	360	50	3

The spike sorting procedure. First, we used *Kilosort 2.0* software to detect and automatically sort spikes. The automatic merging of clusters was based on a spike waveform correlation coefficient criterion of 0.5 (Pachitariu, Sridhar, and Stringer 2023). The automatic splits were based on a bimodality threshold and auto- and cross-correlograms of pairs of clusters (Pachitariu, Sridhar, and Stringer 2023). Second, we used a Python-based GUI (*phy*) to manually verify and sort the output of the *Kilosort* algorithm. We first looked for spike artifacts that appeared in all channels and discarded them. We then looked for spikes that were unstable during a certain duration within a session. If nearby channels had clusters of spikes during those durations, we merged the two clusters of spikes if (i) they had a high correlation of spike waveforms (Pearson's correlation > 0.9) and (ii) they were visually overlapping on the PC space computed over spike waveforms features. We split the spikes by a line on the PC space to maximally separate the two clusters.

We included both single units and multi-units in our analyses. Multi-units refer to those clusters that had no more than two zero-crossings. Since this could also result from the electrode contact being far from the soma, we included multi-units.

Example waveforms, corresponding PSTHs, inter-spike interval (ISI), and auto-correlogram of five example neurons are shown below in Figure R8 as extracted from the open access spike sorting software, *phy*.

Fig R8. Each row shows various cluster statistics for a given neuron and its corresponding PSTH plotted for five inter-landmark distances (left).

References

Gardner, Richard J., Li Lu, Tanja Wernle, May-Britt Moser, and Edvard I. Moser. 2019. "Correlation Structure of Grid Cells Is Preserved during Sleep." *Nature Neuroscience* 22 (4): 598–608.

Pachitariu, Marius, Shashwat Sridhar, and Carsen Stringer. 2023. "Solving the Spike Sorting Problem with Kilosort." *bioRxiv*. <https://doi.org/10.1101/2023.01.07.523036>.

Sargolini, Francesca, Marianne Fyhn, Torkel Hafting, Bruce L. McNaughton, Menno P. Witter, May Britt Moser, and Edvard I. Moser. 2006. "Conjunctive Representation of Position, Direction, and Velocity in Entorhinal Cortex." *Science* 312 (5774): 758–62.

Trettel, Sean G., John B. Trimper, Ernie Hwaun, Ila R. Fiete, and Laura Lee Colgin. 2019. "Grid Cell Co-Activity Patterns during Sleep Reflect Spatial Overlap of Grid Fields during Active Behaviors." *Nature Neuroscience* 22 (4): 609–17.

Reviewer Reports on the First Revision:

Referees' comments:

Referee #1 (Remarks to the Author):

To paraphrase Carl Sagan, extraordinary claims require extraordinary evidence. In this manuscript, the authors are making an extraordinary claim – that grid cells in the primate entorhinal cortex support ‘mental navigation’ to an unmarked location in a virtual 1D environment. But by their own admission, they do not have the requisite evidence to support their claim. They have presented a series of additional analyses to demonstrate that their results *could* be accounted for by grid cells in an attractor network but cannot show that their results are *most likely* accounted for by grid cells in an attractor network. There are a wide range of alternative, much simpler explanations for these findings – including landmark cell responses that have been identified in the entorhinal cortex across a range of mammalian species (as stated previously). In their own words, “landmark cells and grid cells are not clearly dissociable in [this] experiment with regular landmarks”.

To support their conclusion that these results arise from grid cells, they make two key claims: first, that these firing patterns persist when the visual landmarks are removed; and second, that these firing patterns show properties consistent with attractor dynamics. However, it has been well established that landmark (or ‘object trace’ or ‘object vector’) cell responses can persist in the absence of external visual input (e.g. Tsao et al., *Current Biology* 2013; Qasim et al., *Nature Neuroscience* 2019; Poulter et al., *Nature Neuroscience* 2021); and that periodic firing patterns can be generated by alternative network mechanisms (e.g. Burgess et al., *Hippocampus* 2007). Moreover, the fact that the periodicity of these responses exactly matches the temporal offset between landmarks, and that the distribution of relative firing phases for both animals is strongly clustered, argue against these firing patterns arising from grid cells. However the authors wish to interpret their results, there is still a clear discrepancy between the cross-correlation structure in their attractor network model with landmark cell inputs (shown in Fig R3 but - notably - not in the manuscript) and in their data (particularly from Monkey M, shown in Fig S7).

I do not wish to be overly negative – I appreciate the additional analyses and text that the authors have incorporated into their manuscript in response to my original comments, and I don't want to stand in the way of this manuscript being published. As I said in my previous review, these experiments address a topic of considerable current interest (the role of entorhinal cortex in mental navigation), the data are unique, and the combination of modelling and empirical work is to be commended. The presence of periodic firing patterns could be of significant interest, particularly given that those firing patterns can be linked directly with behaviour. I just think that the authors need to be clearer about the limitations of their data in supporting their conclusions. In short, they simply do not know if these responses arise from grid cells, they are just as likely (if not more likely) to arise from landmark cells (for the reasons given above), and that should be clearly stated at some point in the manuscript. In particular, the authors should explicitly compare the cross-correlation structure from their model with their data, analysing and plotting the results in the same way for each, and comment on the discrepancy.

As an aside, I do not think the additional analyses shown in Fig R1 / S10 contribute anything to the manuscript, and I would be minded to remove them – but I am happy to leave that to the authors discretion.

Referee #4 (Remarks to the Author):

The authors report an interesting class of neurons in the macaque entorhinal cortex that appear to be involved in mental navigation. Two monkeys learned a sequence of images by navigating to target images for a reward by using a joystick. After training, they were able to successfully navigate even when the images were no longer visible. Importantly, they were also able to mentally navigate to images from the sequence that they were never trained to navigate toward. This suggests that they learned a mental representation of the entire sequence which is interpreted by the authors as a cognitive map.

A subset of entorhinal neurons had temporally periodic firing that aligned quite precisely with the timing of the invisible landmarks. Another smaller subset of neurons displayed ramping of activity during mental navigation. The periodic cells in particular appear to correlate with trial-to-trial variability such that faster (slower) timescales cause undershooting (overshooting) of behavioral responses.

The study is original and addresses a topic of widespread interest. The data, methodology, analysis, and statistics are solid. The interplay of experimental, theoretical, and modeling work is powerful. The writing is clear and the referencing is appropriate. The conclusions and interpretations are generally valid.

Major comments:

1. The authors argue quite heavily in favor of the hypothesis that the periodic neurons are grid cells. The argument against these neurons being landmark cells on p.21-22 is thorough and logically sound. They need to provide an equally thoughtful discussion of the evidence against these neurons being grid cells. It should also be considered that these periodic neurons do not cleanly map onto any known cell type. The following are perhaps the strongest pieces of evidence against these being grid cells:

- There is extreme clustering of temporal phase whereas grid cells are predicted to have a uniform phase distribution. The authors would like to argue that the clustering is due to plasticity with landmark cells, but this is a prediction of their model, not known data. The phase distributions of the experimental data should be quantified and presented side-by-side with analysis of the model data. Alternative explanations should be carefully considered.
- Their own model suggests that periodicity co-evolves with learning of temporal structure via Hebbian plasticity. This goes against most current theoretical and experimental evidence of grid cells. For example, grid cell spacing is fixed relative to the size of the environment. Grid cell firing

patterns also seem to rapidly map onto novel environments rather than undergoing slow synaptic modifications to learn each new environment. Most importantly, their model predicts that irregular spacing between landmarks would lead to irregular spacing of the neural firing fields. Unfortunately, they do not have experimental data under those conditions, but this result would be directly against the notion of periodic (grid-like) firing patterns that are used as a metric for navigation.

2. Details about corrective behavior are missing on p.3 (and associated methods): “a reward is provided if it matches the target landmark. If not, the animal is given a second and final chance to make a corrective movement and receive a smaller reward (see Methods).” Does this mean the trials classified as correct are actually two populations of responses, both first attempt successes and second attempt successes? This needs to be made explicit. What percentage of trials require a correction? Is the response analyzed the original vector, the correction vector, or the resultant vector after correction? The behavioral and neural responses must be quite different on these error trials and should be analyzed independently somewhere in the manuscript. Justification for pooling first attempt successes and second attempt successes is needed.

3. The authors claim the periodic activity was specific to mental navigation because the periodicity at a 0.65 sec period was reduced during the intertrial interval (ITI) compared to the mental navigation epoch (Fig. S3b). The ITI, however, was only 500-1000 ms. This interval is too short to identify periodicity at 650 ms. They acknowledge this is the case in Fig. S5: “We focused our analysis on trials with a distance of 3, which were long enough to quantify periodicity (i.e. two full periods).” Given that they can only detect periodicity up to a maximum period of 250-500 ms (depending on the trial), this result should be removed and claims appropriately adjusted. The trial-to-trial variability in periodicity described immediately after (Fig. S3c) provides a nice link to behavior that is well justified.

Minor comments:

1. Credit of previous work is lacking on p.3 and p.30. Horner et al. (2016) (<https://doi.org/10.1016/j.cub.2016.01.042>) and Bellmund et al. (2016) (<https://doi.org/10.7554/eLife.17089>) report grid-like activity in humans during mental navigation and imagination. While the work here provides novel evidence from single unit recordings, these papers need to be cited and the wording needs to be modified.

- p.3 “A critical but ****untested**** prediction of the cognitive map hypothesis is that the brain can exploit the structure of the latent map in the absence of sensory inputs to perform purely mental computations.”

- p.30 “Our results provide compelling evidence for the recruitment of a cognitive map in EC during mental navigation.”

2. One of the most similar past findings comes from single neuron recordings in human entorhinal cortex where they find temporally periodic activity during continuous movie watching (Aghajan, Kreiman, & Fried, 2023; <https://doi.org/10.1016/j.celrep.2023.113271>). The authors of that study do not interpret these to be grid cells, instead suggesting they are a complementary cell population that

preferentially encodes metric time over metric space. This article should be discussed, or at least cited.

3. In Fig. S7, S8, and S10 the axis labels and legends report “firing rate cross-correlation averaged over a lag window of ± 5 ms.” Is this meant to say 500 ms? That would better match the plots showing cross correlation structure over lags of ± 650 ms. If it truly is only 5 ms, this warrants a detailed explanation and the results should be recalculated at much longer time lags.

4. All figure legends should be double-checked to ensure completeness. For example, what do the shaded areas in Fig. 2a represent? What are the red lines in Fig. 2b?

5. References are lacking on p.15, paragraphs 1 and 2. The distinctions between the offset coding described here and classic timing studies are unclear without appropriate references for the following statements:

- "Ramping activity has long been associated with timing. In general, three aspects of the ramp can encode a target time interval: the ramp's initial state, its slope, and its end state."
- "The presence of strong offset coding in EC differs qualitatively from ramping activity in other brain areas associated with classical time interval production tasks. In timing tasks that do not involve mental navigation, the target interval is usually encoded by the initial state and the slope, and not by the end state when ramping activity reaches a common threshold."
- "This result further highlights the distinct signatures of mental navigation through time compared to classical timing tasks in which adjustments of slope play a central role."

Author Rebuttals to First Revision:

We thank the reviewers for their careful reading of our manuscript and insightful comments. Here, we provide an overview of how we have addressed the comments.

Referee #1 (Remarks to the Author):

*To paraphrase Carl Sagan, extraordinary claims require extraordinary evidence. In this manuscript, the authors are making an extraordinary claim – that grid cells in the primate entorhinal cortex support ‘mental navigation’ to an unmarked location in a virtual 1D environment. But by their own admission, they do not have the requisite evidence to support their claim. They have presented a series of additional analyses to demonstrate that their results *could* be accounted for by grid cells in an attractor network but cannot show that their results are *most likely* accounted for by grid cells in an attractor network. There are a wide range of alternative, much simpler explanations for these findings – including landmark cell responses that have been identified in the entorhinal cortex across a range of mammalian species (as stated previously). In their own words, “landmark cells and grid cells are not clearly dissociable in [this] experiment with regular landmarks”.*

To support their conclusion that these results arise from grid cells, they make two key claims: first, that these firing patterns persist when the visual landmarks are removed; and second, that these firing patterns show properties consistent with attractor dynamics. However, it has been well established that landmark (or ‘object trace’ or ‘object vector’) cell responses can persist in the absence of external visual input (e.g. Tsao et al., Current Biology 2013; Qasim et al., Nature Neuroscience 2019; Poulter et al., Nature Neuroscience 2021); and that periodic firing patterns can be generated by alternative network mechanisms (e.g. Burgess et al., Hippocampus 2007). Moreover, the fact that the periodicity of these responses exactly matches the temporal offset between landmarks, and that the distribution of relative firing phases for both animals is strongly clustered, argue against these firing patterns arising from grid cells. However the authors wish to interpret their results, there is still a clear discrepancy between the cross-correlation structure in their attractor network model with landmark cell inputs (shown in Fig R3 but - notably - not in the manuscript) and in their data (particularly from Monkey M, shown in Fig S7).

I do not wish to be overly negative – I appreciate the additional analyses and text that the authors have incorporated into their manuscript in response to my original comments, and I don't want to stand in the way of this manuscript being published. As I said in my previous review, these experiments address a topic of considerable current interest (the role of entorhinal cortex in mental navigation), the data are unique, and the combination of modelling and empirical work is to be commended. The presence of periodic firing patterns could be of significant interest, particularly given that those firing patterns can be linked directly with behaviour.

I just think that the authors need to be clearer about the limitations of their data in supporting their conclusions. In short, they simply do not know if these responses arise from grid cells, they are just as likely (if not more likely) to arise from landmark cells (for the reasons given above), and that should be clearly stated at some point in the manuscript.

Thank you for your favorable review despite the difficulty of conclusively knowing whether the cells we report are grid cells or landmark-trace cells. We have altered the text throughout de-emphasizing the focus on grid cells. Moreover, we have added an entire paragraph discussing in detail the evidence for and against the possibility

that the periodic neurons are landmark cells, grid cells or a different type of cell. We have included some excerpts here for convenience:

First, we consider landmark cells. *“the periodic neurons could be homologous to landmark cells.”* We then detail the evidence for and against this possibility and conclude that *“Together, these results are inconsistent with the interpretation that our task-modulated neurons are canonical landmark cells. However, we know nearly nothing about landmark cells in the primate brain. It is therefore possible that the EC neurons we have identified function as landmark cells with the additional capacity to express attractor dynamics.”* Next we consider grid cells. *“Second, our periodic task-modulated neurons may be homologous to GC cells in rodents”.* We next detail the evidence for and against the possibility and conclude *“endogenous periodicity and the preserved cell-cell correlations are consistent with this interpretation. However, we found a significantly tighter clustering of the relative phase distribution of cell-cell cross-correlation in our data, which differs from what is predicted for GC in rodents. This clustering may be partially due to plasticity of connections to putative landmark cells but the clustering was stronger in our data compared to the GC model units (Fig S8). These discrepancies suggest that either the periodic neurons are not homologous to GC or that the attractor network supporting grid-like activity in primates differs from rodents.”* Finally, we consider the possibility that *“neurons in our population are neither landmark cells nor GCs but are parts of an attractor network capable of producing memory traces for the landmarks within the functional architecture of entorhinal cortex^{29–31}.”*

In particular, the authors should explicitly compare the cross-correlation structure from their model with their data, analysing and plotting the results in the same way for each, and comment on the discrepancy.

As an aside, I do not think the additional analyses shown in Fig R1 / S10 contribute anything to the manuscript, and I would be minded to remove them – but I am happy to leave that to the authors discretion.

As the reviewer has suggested, we compared the distribution of relative phases across units in the CAN model and in the neural data from both monkeys. We have added these new comparative plots as Fig S8. As surmised by the reviewer, this result suggests that the periodic neurons might not be homologous to the canonical grid cells found in rodents. We have highlighted this discrepancy and used it as evidence against the interpretation that our data reflect grid cells. We are grateful for this suggestion. While our data and the presence of attractor dynamics generating endogenous periodicity is clear, we think the new analyses provide a much more balanced interpretation of the data.

The revised text includes *“However, we found a significantly tighter clustering of the relative phase distribution of cell-cell cross-correlation in our data, which differs from what is predicted for GC in rodents. This clustering may be partially due to plasticity of connections to putative landmark cells but the clustering was stronger in our data compared to the GC model units (Fig S8). These discrepancies suggest that either the periodic neurons are not homologous to GC or that the attractor network supporting grid-like activity in primates differs from rodents.”*

Referee #4 (Remarks to the Author):

The authors report an interesting class of neurons in the macaque entorhinal cortex that appear to be involved in mental navigation. Two monkeys learned a sequence of images by navigating to target images for a reward by using a joystick. After training, they were able to successfully navigate even when the images were no longer visible. Importantly, they were also able to mentally navigate to images from the sequence that they were never trained to navigate toward. This suggests that they learned a mental representation of the entire sequence which is interpreted by the authors as a cognitive map.

A subset of entorhinal neurons had temporally periodic firing that aligned quite precisely with the timing of the invisible landmarks. Another smaller subset of neurons displayed ramping of activity during mental navigation. The periodic cells in particular appear to correlate with trial-to-trial variability such that faster (slower) timescales cause undershooting (overshooting) of behavioral responses.

The study is original and addresses a topic of widespread interest. The data, methodology, analysis, and statistics are solid. The interplay of experimental, theoretical, and modeling work is powerful. The writing is clear and the referencing is appropriate. The conclusions and interpretations are generally valid.

Thank you for your favorable review despite the difficulty of conclusively knowing the class of neurons involved in mental navigation.

Major comments:

1. The authors argue quite heavily in favor of the hypothesis that the periodic neurons are grid cells. The argument against these neurons being landmark cells on p.21-22 is thorough and logically sound. They need to provide an equally thoughtful discussion of the evidence against these neurons being grid cells. It should also be considered that these periodic neurons do not cleanly map onto any known cell type. The following are perhaps the strongest pieces of evidence against these being grid cells:

- There is extreme clustering of temporal phase whereas grid cells are predicted to have a uniform phase distribution. The authors would like to argue that the clustering is due to plasticity with landmark cells, but this is a prediction of their model, not known data. The phase distributions of the experimental data should be quantified and presented side-by-side with analysis of the model data. Alternative explanations should be carefully considered.*

- Their own model suggests that periodicity co-evolves with learning of temporal structure via Hebbian plasticity. This goes against most current theoretical and experimental evidence of grid cells. For example, grid cell spacing is fixed relative to the size of the environment. Grid cell firing patterns also seem to rapidly map onto novel environments rather than undergoing slow synaptic modifications to learn each new environment. Most importantly, their model predicts that irregular spacing between landmarks would lead to irregular spacing of the neural firing fields. Unfortunately, they do not have experimental data under those conditions, but this result would be directly against the notion of periodic (grid-like) firing patterns that are used as a metric for navigation.*

Following the reviewer's suggestion, we have added an entire paragraph discussing in detail the evidence for and against the possibility that the periodic neurons are landmark cells, grid cells or a different type of cell. We have included this paragraph here for convenience:

"In general, we envision three possibilities. First, the periodic neurons could be homologous to landmark cells recorded in rodents during navigation. Previous work suggests that landmark neurons are modulated during the presence of visible landmarks and not when landmarks are invisible³⁹. Accordingly, when we applied our analysis to landmark neurons in a previous study³⁹, cross-correlations were preserved across epochs with visible landmarks but abolished when the landmarks were invisible [STATS]. Furthermore, the preserved correlations during the inference epoch in the MNAV task make it unlikely that these neurons were landmark cells. During this epoch, the animals were presented with the start and target landmarks. If these neurons were landmark cells, we would expect the strong visual inputs from the visible start and target landmarks to weaken correlations. However, we found no such drop in cross-correlations [STATS] without a significant drop in magnitude compared to the navigation epoch [STATS]. Together, these results are inconsistent with the interpretation that our task-modulated neurons are canonical landmark cells. However, we know nearly nothing about landmark cells in the primate brain. It is therefore possible that the EC neurons we have identified function as landmark cells with the additional capacity to express attractor dynamics. Second, our periodic task-modulated neurons may be homologous to GC cells in rodents. The presence of endogenous periodicity and the preserved cell-cell correlations are consistent with this interpretation. However, we found a significantly tighter clustering of the relative phase distribution of cell-cell cross-correlation in our data, which differs from what is predicted for GC in rodents. This clustering may be partially due to the plasticity of connections to putative landmark cells, but the clustering was stronger in our data compared to the GC model units (Fig S8). Moreover, many studies have found that the firing patterns of GCs can remap rapidly in response to both environmental factors⁶⁸⁻⁷⁰ and internal variables⁷¹ without the need for slow synaptic modifications. Yet, other studies have found evidence for slow learning within the grid system^{27,72}, as predicted by our model. Therefore, at this stage, we cannot make a definitive statement about whether GCs undergo slow synaptic modifications during mental navigation. One experimental modification that could shed light on this question is to record from this cell population in a variant of our task with irregular spacing between landmarks. In that scenario, we would be able to identify these neurons as GC more definitively if they maintain their periodicity. Therefore, we conclude that either the periodic neurons in our dataset are not homologous to GC or that the attractor network supporting grid-like activity in primates differs from rodents. A final alternative is that the periodic neurons in our population are neither landmark cells nor GCs but are part of an attractor network within the functional architecture of the entorhinal cortex that is capable of producing memory traces for the landmarks²⁹⁻³¹."

2. Details about corrective behavior are missing on p.3 (and associated methods): "a reward is provided if it matches the target landmark. If not, the animal is given a second and final chance to make a corrective movement and receive a smaller reward (see Methods)." Does this mean the trials classified as correct are actually two populations of responses, both first attempt successes and second attempt successes? This needs to be made explicit. What percentage of trials require a correction? Is the response analyzed the original vector, the correction vector, or the resultant vector after correction? The behavioral and neural responses must be quite different on these error trials and should be analyzed independently somewhere in the manuscript. Justification for pooling first attempt successes and second attempt successes is needed.

Thank you for pointing out this source of confusion. Only single-attempt trials were considered as correct trials. Therefore, correct trials are not two populations of responses. Only in Supp fig S3c, we considered incorrect trials. We have clarified this in pg3 and Methods with the following text:

“Throughout the paper, only single-attempt trials are considered as correct trials. In Fig S3c, we considered incorrect trials, which are defined as trials in which animals made more than one attempt regardless of whether the animal completed the trial in its second attempt. For the analysis in S3c, we obtained the data from the first attempt of incorrect trials.”

3. The authors claim the periodic activity was specific to mental navigation because the periodicity at a 0.65 sec period was reduced during the intertrial interval (ITI) compared to the mental navigation epoch (Fig. S3b). The ITI, however, was only 500-1000 ms. This interval is too short to identify periodicity at 650 ms. They acknowledge this is the case in Fig. S5: “We focused our analysis on trials with a distance of 3, which were long enough to quantify periodicity (i.e. two full periods).” Given that they can only detect periodicity up to a maximum period of 250-500 ms (depending on the trial), this result should be removed and claims appropriately adjusted. The trial-to-trial variability in periodicity described immediately after (Fig. S3c) provides a nice link to behavior that is well justified.

Thank you for pointing out another source of confusion. Although the ITI was set to be 500-1000ms, each trial only started after animals acquired fixation for 200ms. As such, we were able to grab several epochs during the experiment in between trials when the ITI was long. We therefore pooled segments of ITI data longer than 4sec throughout a session to generate the comparison scatter plots in Fig S3b. We have added a section in the Methods to clarify this point.

“Since each trial only started after animals acquired fixation for 200ms, we were able to acquire several ITI segments longer than 4 seconds. We pooled these segments of data for comparative analyses (e.g. Fig S3b and 3e).”

Figure S5 was a special case in which we were comparing the top and bottom tertiles of the trials sorted by timing error. Therefore, we focused our analysis on distance 3. While distances of 4 and 5 didn't afford statistical power for this particular analysis, the rest of the analyses in the paper take into account distances greater than or equal to 3.

Minor comments:

1. Credit of previous work is lacking on p.3 and p.30. Horner et al. (2016) (<https://doi.org/10.1016/j.cub.2016.01.042>) and Bellmund et al. (2016) (<https://doi.org/10.7554/eLife.17089>) report grid-like activity in humans during mental navigation and imagination. While the work here provides novel evidence from single unit recordings, these papers need to be cited and the wording needs to be modified.

*• p.3 “A critical but ****untested**** prediction of the cognitive map hypothesis is that the brain can exploit the structure of the latent map in the absence of sensory inputs to perform purely mental computations.”*

• p.30 “Our results provide compelling evidence for the recruitment of a cognitive map in EC during mental navigation.”

Thank you! We have now cited the suggested papers and altered the text on pg3 and pg30.

2. One of the most similar past findings comes from single neuron recordings in human entorhinal cortex where they find temporally periodic activity during continuous movie watching (Aghajan, Kreiman, & Fried, 2023; <https://doi.org/10.1016/j.celrep.2023.113271>). The authors of that study do not interpret these to be grid cells, instead suggesting they are a complementary cell population that preferentially encodes metric time over metric space. This article should be discussed, or at least cited.

Thank you! We have now cited the suggested paper and added text to the Discussion as follows:

“Alternatively, the periodic neurons could be of the same class of neurons recently discovered in rodent MEC (Gonzalo Cogno et al. 2023) and human EC (M Aghajan et al. 2023) exhibiting periodicity ranging from seconds to minutes. However, without rigorous behaviorally controlled studies, it is difficult to conclude if these newly found periodic neurons are all the same cell type or all serve the same functional purpose.”

3. In Fig. S7, S8, and S10 the axis labels and legends report “firing rate cross-correlation averaged over a lag window of +/-5ms.” Is this meant to say 500 ms? That would better match the plots showing cross correlation structure over lags of +/- 650 ms. If it truly is only 5 ms, this warrants a detailed explanation and the results should be recalculated at much longer time lags.

The goal of these scatter plot analyses is to quantify the similar patterns of pairwise cross-correlograms across epochs. So, we averaged the cross-correlograms within a ± 5 ms interval around zero to obtain a single short-latency cross-correlation estimate for each grid-cell pair, as done in previous studies (Trettel et al Nat Neuro 2019; Gardner et al Nat Neuro 2019). We now repeat this analysis at all time lags over +650ms as the reviewer suggested. We have added this result in panel c of the modified Fig S7.

c, Correlation between cross-correlation values during mental navigation epoch vs ITI (left) and vs inference epoch (right) at various lags (left). Left panels: monkey A, right panels: Monkey M. Red stars denote the significance of Pearson’s correlation at $p < .05$ with Bonferroni correction.

4. All figure legends should be double-checked to ensure completeness. For example, what do the shaded areas in Fig. 2a represent? What are the red lines in Fig. 2b?

Thank you for your suggestion. We have now added clarifying text to the legend of Figures 2a and 2b.

5. References are lacking on p.15, paragraphs 1 and 2. The distinctions between the offset coding described here and classic timing studies are unclear without appropriate references for the following statements:

- "Ramping activity has long been associated with timing. In general, three aspects of the ramp can encode a target time interval: the ramp's initial state, its slope, and its end state."*
- "The presence of strong offset coding in EC differs qualitatively from ramping activity in other brain areas associated with classical time interval production tasks. In timing tasks that do not involve mental navigation, the target interval is usually encoded by the initial state and the slope, and not by the end state when ramping activity reaches a common threshold."*
- "This result further highlights the distinct signatures of mental navigation through time compared to classical timing tasks in which adjustments of slope play a central role."*

Thank you for your suggestion. We have now cited several old and recent studies.

Reviewer Reports on the Second Revision:

Referees' comments:

Referee #1 (Remarks to the Author):

I am satisfied that the authors have addressed my previous concerns, and that the manuscript now provides a more balanced reflection of their results. One final comment - I think the new section of text that compares and contrasts landmark and grid cell interpretations of these data could be significantly compressed, and / or split into more than one paragraph, and / or some of the details integrated into the Discussion. I would also caution against any statistical result being given as $p=1$

Referee #4 (Remarks to the Author):

The authors have successfully addressed all of my concerns. In particular, I appreciate their willingness to provide a more balanced discussion of which entorhinal cell types may be recruited in their task.